# Pt size-dependent reverse oxygen spillover on Sn-doped Pt/TiO$_2$ for CO oxidation

Shangchao Xiong [1,2,3,4], Zhengjun Gong [2,4], Houlin Wang[1,2], Jianqiang Shi[1], Haiyan Liu[1], Xiaoping Chen[1], Jinxing Mi[1], Jianjun Chen [1] ✉ & Junhua Li [1]

Interfacial oxygen migration from support to noble metal active sites, termed reverse oxygen spillover, represents a critical metal-support interaction influencing the performance of Pt/TiO$_2$ catalysts. In this study, we uncover a size effect of Pt particles on reverse oxygen spillover in Pt/Sn$_{0.2}$Ti$_{0.8}$O$_2$ catalysts via a combination of in situ characterizations with ab initio molecular dynamics simulations. Among single-atom Pt, nanocluster Pt, and nanocrystal Pt, nanocluster Pt exhibits the most pronounced reverse oxygen spillover and thus achieves the highest turnover frequency in CO oxidation. The most pronounced reverse oxygen spillover is mainly due to the strongest electron transfer to the interfacial lattice oxygen triggered by CO adsorption with moderate adsorption energy. In contrast, CO adsorption on single-atom Pt is too strong to initiate reverse oxygen spillover, while on nanocrystal Pt, it leads to a weakening of the interaction between Pt sites and the support, thus hinders the reverse oxygen spillover. This study clarifies the relationship between Pt particle size and reverse oxygen spillover effects, furnishing a theoretical basis for designing noble metal catalysts with excellent activity.

Supported noble metal catalysts play an important role to new technologies for both energy and environmental applications[1–5]. Platinum (Pt), as a noble metal, exhibits outstanding catalytic performance in many redox reactions[6–8]. When supported by TiO$_2$, Pt exhibits variable metal-support interactions (MSI) and synergistic effects, making Pt/TiO$_2$ catalysts highly promising in diverse areas, including energy conversion, environmental protection, and chemical synthesis[9,10]. Therefore, the understanding of fundamental principles in MSI phenomena is of great importance to the rational design of Pt/TiO$_2$ catalysts.

Within the realm of metal-support interaction (MSI), the migration of substances at the metal/support interface can significantly influence the catalytic performance[11–13]. One type of the substance migration is mass transfer from the support to Pt, which has been relatively thoroughly investigated in the case of anatase TiO$_2$ support.

Upon a reduction treatment, the partially reduced anatase TiO$_2$ migrates onto the Pt active phase, leading to a partial encapsulation of the Pt particles[14,15]. Given that such mass transfer results in the encapsulation or partial encapsulation of the Pt active phase, it generally exerts an adverse effect on the catalytic activity. Another form of substance migration is the transfer of O species, known as the reverse oxygen spillover (ROS)[16–18], which is believed to be of great significance for the formation of catalytic indispensable Pt-O species. However, as a complex interfacial phenomenon, the investigation on ROS is highly challenging. Thermodynamically, ROS requires that the adsorption energy of oxygen on Pt be lower than the formation energy of oxygen vacancies on the support, making this phenomenon not exist on all metal/support systems. Kinetically, ROS occurs on a picosecond timescale, making it difficult to observe. In our previous work[19], we doped Sn into rutile TiO$_2$ supports to enhance the mobility of lattice

[1]State Key Joint Laboratory of Environment Simulation and Pollution Control, School of Environment, Tsinghua University, Beijing, China. [2]Sichuan International Science and Technology Cooperation base for Intelligent Environmental Protection and Sustainable Development in Rail Transit, School of Environmental Science and Engineering, Southwest Jiaotong University, Chengdu, China. [3]Department of Civil and Environmental Engineering, The Hong Kong Polytechnic University, Hung Hom, Kowloon, Hong Kong, China. [4]These authors contributed equally: Shangchao Xiong, Zhengjun Gong. ✉e-mail: chenjianjun@tsinghua.edu.cn

oxygen by creating asymmetric oxygen sites, and depicted that ROS is triggered by CO adsorption on $Pt^{2+}$ followed by bond cleavage of asymmetric Ti-O-Sn moieties and the appearance of $Pt^{4+}$ species. The ROS enables catalyst with superior CO oxidation activity and high sulfur resistance.

The size effect of Pt particles also has a significant impact on catalytic reactions, as it can alter the dispersion, physical and electronic structure of Pt particles, thereby significantly affecting the adsorption and activation of reactants at the active metal sites[20–23]. The influence of Pt particle size on catalytic activity is multifaceted, and the key factors exhibit uncertainty across different reactions, making it both challenging and meaningful to determine the optimal metal particle size for Pt-based catalysts. Specifically, when supported on reducible oxide, Pt size could influence surface properties of support due to modification of interaction between metal and support. For example, it is found that Pt size could adjust the redox properties and $O_2$-activation of $Pt/Fe_2O_3$ catalysts[24]. On $Pt/CeO_2$ catalysts, the Pt size influences the mobility of lattice oxygen of $CeO_2$ and consequently affect the activity of CO oxidation[25–27]. Though previous studies give a good understanding for the size effect of Pt particle supported on active support, the dependence of Pt particle size on the ROS is never investigated.

In this paper, Sn was doped into $TiO_2$ to synthesize a Ti-based support with asymmetric interfacial oxygen sites. The incorporation of Sn facilitated the phase transformation of $TiO_2$ into the thermodynamically more stable rutile structure. Pt was then loaded onto the support at varying weights to create a series of Pt-based catalyst systems ranging from single-atom to nanocluster and nanocrystal. A combination of in situ experimental characterizations and ab initio molecular dynamics (AIMD) simulations was employed to investigate the influence of Pt particle size on the ROS process and its effect on CO catalytic oxidation. The results revealed that single-atom, nanocluster, and nanocrystal Pt exhibit fundamentally different impacts on ROS. Only nanocluster Pt demonstrated optimal CO adsorption and a significant accumulation of electrons around the interfacial oxygen, facilitating the migration of interfacial oxygen towards the Pt active site. Consequently, nanocluster Pt exhibited a pronounced ROS effect, enhancing CO catalytic oxidation and achieving a turnover frequency (TOF) approximately twice that of the single-atom and nanocrystal Pt. These findings provide deeper insight into the interfacial oxygen migration mechanism and offer a theoretical basis for designing catalysts with ROS-enhanced activity for CO removal in industrial furnace flue gas.

## Results

### Structural analysis

As shown in Fig. 1a, $Sn_{0.2}Ti_{0.8}O_2$ (denoted as STO) support was prepared via a coprecipitation method, followed by the deposition of 0.1, 0.25, 0.5, and 1.0 wt% Pt on the STO support and a $H_2$ pretreatment process to prepare a series of Pt/STO catalysts (denoted as 0.1Pt/STO,

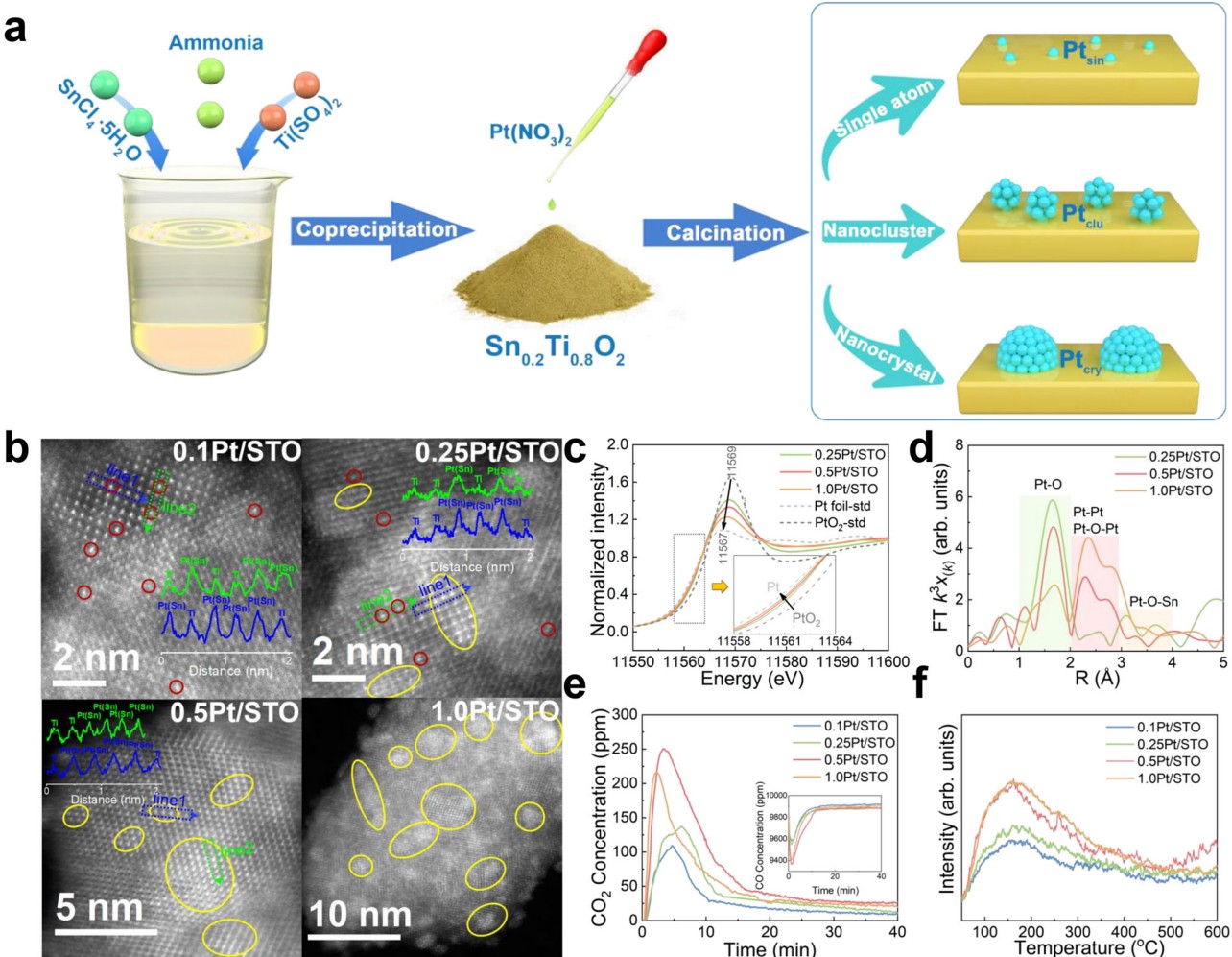

**Fig. 1 | Structural characterization of Pt/STO catalysts. a** Schematic diagram of the Pt/STO catalysts synthesis process. **b** AC-HAADF-STEM images; the inset shows intensity analysis of the corresponding-colored region; red circle indicates single-atom Pt, yellow circle indicates nanocluster and nanocrystal Pt. **c** Pt $L_3$-edge XANES spectra. **d** $k^3$-weighted FT-EXAFS of Pt $L_3$-edge. **e** Transient CO and $CO_2$ concentration changes during 1% $CO/N_2$ flow at 100 °C. **f** $O_2$-TPD profile.

0.25Pt/STO, 0.5Pt/STO, and 1.0Pt/STO). The XRD patterns of all Pt/STO catalysts (Supplementary Fig. S1) perfectly match that of rutile $TiO_2$ (JCPDS: #21–1276), with no peaks attributable to $PtO_x$ or $SnO_x$. The main diffraction peak at 27.2° shows a slight shift compared with the standard pattern of $TiO_2$ (27.3°), indicating the successful doping of Sn into the $TiO_2$ lattice. Moreover, the main peak position and half-peak width of all Pt/STO catalysts remain essentially unchanged, confirming that Pt loading has little influence on the crystal structure of the STO support. The $N_2$ adsorption-desorption curves (Supplementary Fig. S2a) and pore size distribution plots (Supplementary Fig. S2b) of all Pt/STO catalysts are nearly identical, indicating that varying Pt content does not significantly affect the BET specific surface area, total pore volume, or average pore width (Supplementary Table S1). The aberration-corrected high-angle annular dark-field scanning transmission electron microscopy (AC-HAADF-STEM) results (Supplementary Figs. S3 and S4) demonstrate that both crystalline Pt and the STO support predominantly expose the (110) crystal plane (see Supplementary Note 1 for analysis).

As shown in the AC-HAADF-STEM images (Fig. 1b), Pt species evolve with increasing loading: 0.1Pt/STO contains mainly single atoms; 0.25Pt/STO features both single atoms and nanoclusters; 0.5Pt/STO is dominated by nanoclusters; and 1.0Pt/STO exhibits aggregated nanocrystals. Additionally, the intensity analysis of AC-STEM HAADF images reveals a high degree of overlap between Pt and Sn sites, indicating that Pt has a preferential affinity to Sn sites. Supplementary Fig. S5 shows the AC-STEM EDS elemental mapping of Pt/STO catalysts, which confirms that the O, Ti, and Sn elements are evenly distributed, providing further evidence that Sn has been successfully incorporated into the $TiO_2$ lattice. The elemental mapping of Pt reveals that Pt aggregates as its loading increases, consistent with the findings in Fig. 1b.

The microstructural characteristics of Pt was further analyzed using X-ray absorption near-edge spectroscopy (XANES). Given the low Pt content in 0.1Pt/STO, the signal-to-noise ratio of the XANES data was inadequate. Consequently, a comprehensive analysis was performed exclusively on the 0.25Pt/STO, 0.5Pt/STO, and 1.0Pt/STO catalysts. As shown in Fig. 1c, the absorption edge of the Pt $L_3$-edge XANES spectra for all Pt/STO catalysts lie between those of the Pt foil and $PtO_2$ standard (Pt foil-std and $PtO_2$-std) samples, indicating that the Pt in the Pt/STO catalysts exists in the form of $PtO_x$. Additionally, as the Pt loading increases, the white line intensity of the Pt/STO catalysts slightly decreases, suggesting that an elevation in Pt loading leads to a modest decrease in the average oxidation state of Pt[14,28]. The coordination state of Pt was further analyzed by Fourier-transform extended X-ray absorption fine structure (FT-EXAFS), with detailed fitting results in Supplementary Fig. S6 and Supplementary Table S2. Figure 1d shows a visual comparison of the FT-EXAFS spectra for Pt/STO catalysts. A distinct peak is observed around 2 Å, which corresponds to the Pt–O bond, confirming that Pt is in the $PtO_x$ state[14,29]. As the Pt loading increases, a notable decrease in the intensity of the Pt–O bond is observed, while the intensity of peaks corresponding to Pt–Pt and Pt–O–Pt increases markedly. This suggests that higher Pt loading leads to significant Pt aggregation, a conclusion consistent with the results in Fig. 1b. Furthermore, coordination peaks attributable to Pt–O–Sn were observed in the Pt/STO catalysts (Fig. 1d and Supplementary Table S2), indicating that Pt tends to be coordinated with Sn sites, which is also in agreement with the AC-HAADF-STEM results.

It is widely acknowledged that the oxidation of CO over Pt-based catalysts proceeds according to the Mars–van Krevelen (MvK) mechanism. In this process, CO initially adsorbs onto Pt sites and is subsequently oxidized by active lattice oxygen to produce $CO_2$[30–32]. Therefore, in transition CO oxidation reaction without $O_2$, the amount of consumed CO and produced $CO_2$ can serve as reliable indicators for estimating the quantity of active lattice oxygen on Pt-based catalysts. The experimental results (Fig. 1e) indicate that the trend in active lattice oxygen content is as follows: 0.5Pt/STO > 1.0Pt/STO > 0.25Pt/STO > 0.1Pt/STO. To further verify this conclusion, all Pt/STO catalysts were treated with $^{18}O_2$ at 500 °C for 1 h to partially substitute the surface active oxygen species with $^{18}O$. Subsequently, the transition CO oxidation reaction was carried out again at 100 °C. As shown in Supplementary Fig. S7, the formation sequence of $C^{16}O^{16}O$ and $C^{16}O^{18}O$ is consistent with the above conclusion, while almost no $C^{18}O^{18}O$ was detected. Interestingly, the $O_2$ temperature-programmed desorption ($O_2$-TPD) experiment (Fig. 1f) reveals that the content of $O_2$ desorption on 1.0Pt/STO is slightly higher than on 0.5Pt/STO. The discrepancy between the transition CO oxidation and $O_2$-TPD result can be attributed to the fact that a portion of the active oxygen species presented on the surface of the Pt/STO catalyst are generated in situ during the CO oxidation reaction, and this in situ-generated oxygen cannot be characterized by $O_2$-TPD technique. Furthermore, it is plausible to attribute this discrepancy observed among the Pt/STO catalyst to the variation in Pt sizes. Consequently, it is likely that Pt size variation significantly influence the in situ generation of active oxygen. To verify this hypothesis, a comprehensive in situ characterizations were conducted.

## Effect of Pt size on ROS

In situ near-ambient pressure X-ray photoelectron spectroscopy (NAP-XPS) and Raman spectroscopy were used to observe the chemical state changes of Pt/STO catalysts at 100 °C, both for the original samples and under sequences of $O_2$, CO + $O_2$, and CO exposures. Unfortunately, due to the low Pt content in the 0.1Pt/STO catalyst, the signal-to-noise ratio was too low for meaningful analysis of the obtained NAP-XPS data. Additionally, the Ti 3s satellite peak at ~75 eV overlapped significantly with the Pt 4f peaks[14,33,34], which made the Pt 4f spectra of the Pt/STO catalysts particularly complex. To obtain reliable peak-fitting results for Pt 4f in NAP-XPS, the fitting process strictly follows the criteria that area ratio of the Pt $4f_{7/2}$ to $4f_{5/2}$ peaks was fixed at 4:3, and the full width at half maximum (FWHM) for all Pt peaks was kept the same. The resulting data (Fig. 2a–c) indicates peaks at ~77.8 eV and ~74.5 eV corresponding to the $4f_{5/2}$ and $4f_{7/2}$ states of $Pt^{4+}$, while peaks at ~75.5 eV and ~72.3 eV were attributed to $Pt^{2+}$ in the $4f_{5/2}$ and $4f_{7/2}$ states[35,36]. Throughout the NAP-XPS tests, no metallic Pt peaks were observed. After $H_2$ treatment, all Pt/STO catalysts displayed only $Pt^{2+}$ characteristic peaks, indicating that the surface Pt was predominantly in the form of PtO, consistent with the XANES results.

Interestingly, after exposure to $O_2$, the $Pt^{2+}$ species on the Pt/STO catalysts remained stable, and no significant oxidation to $Pt^{4+}$ was observed. However, when both CO and $O_2$ were introduced, a clear $Pt^{4+}$ peak emerged, which persisted even when only CO was introduced. This suggests that $Pt^{2+}$ on the Pt/STO surface is resistant to oxidation by $O_2$. Instead, when a reducing agent CO is introduced, oxygen species migrate from the support to the active Pt sites (reverse O spillover, ROS), thereby oxidizing $Pt^{2+}$ to $Pt^{4+}$. Consequently, the quantity of $Pt^{4+}$ detected in NAP-XPS upon CO and CO + $O_2$ exposure is indicative of the strength of ROS effect. As shown in Fig. 2a–c, the $Pt^{4+}$ contents on 0.5Pt/STO under CO and CO + $O_2$ exposure were notably higher than those on 0.25Pt/STO and 1.0Pt/STO. The AC-HAADF-STEM image (Fig. 1b) reveals that Pt in 0.5Pt/STO is primarily in the form of nanoclusters, while 0.25Pt/STO comprises a combination of single-atom species and nanoclusters, and 1.0Pt/STO features mainly nanocrystals. Therefore, it can be deduced that the particle size of Pt significantly affects the strength of the ROS effect, with nanocluster Pt demonstrating the highest efficiency in promoting this effect.

The in situ Raman spectroscopic analysis provides additional corroboration for the aforementioned findings. As shown in Fig. 2d–g, all the pristine Pt/STO catalysts exhibited peaks at approximately 600, 411, and 252 $cm^{-1}$, corresponding to the $A_{1g}$ vibration, $E_g$ vibration, and second-order effect (SOE) of rutile $TiO_2$[37,38], respectively. For 0.25Pt/STO, 0.5Pt/STO and 1.0Pt/STO catalysts, the $A_{1g}$ and $E_g$ vibration peaks

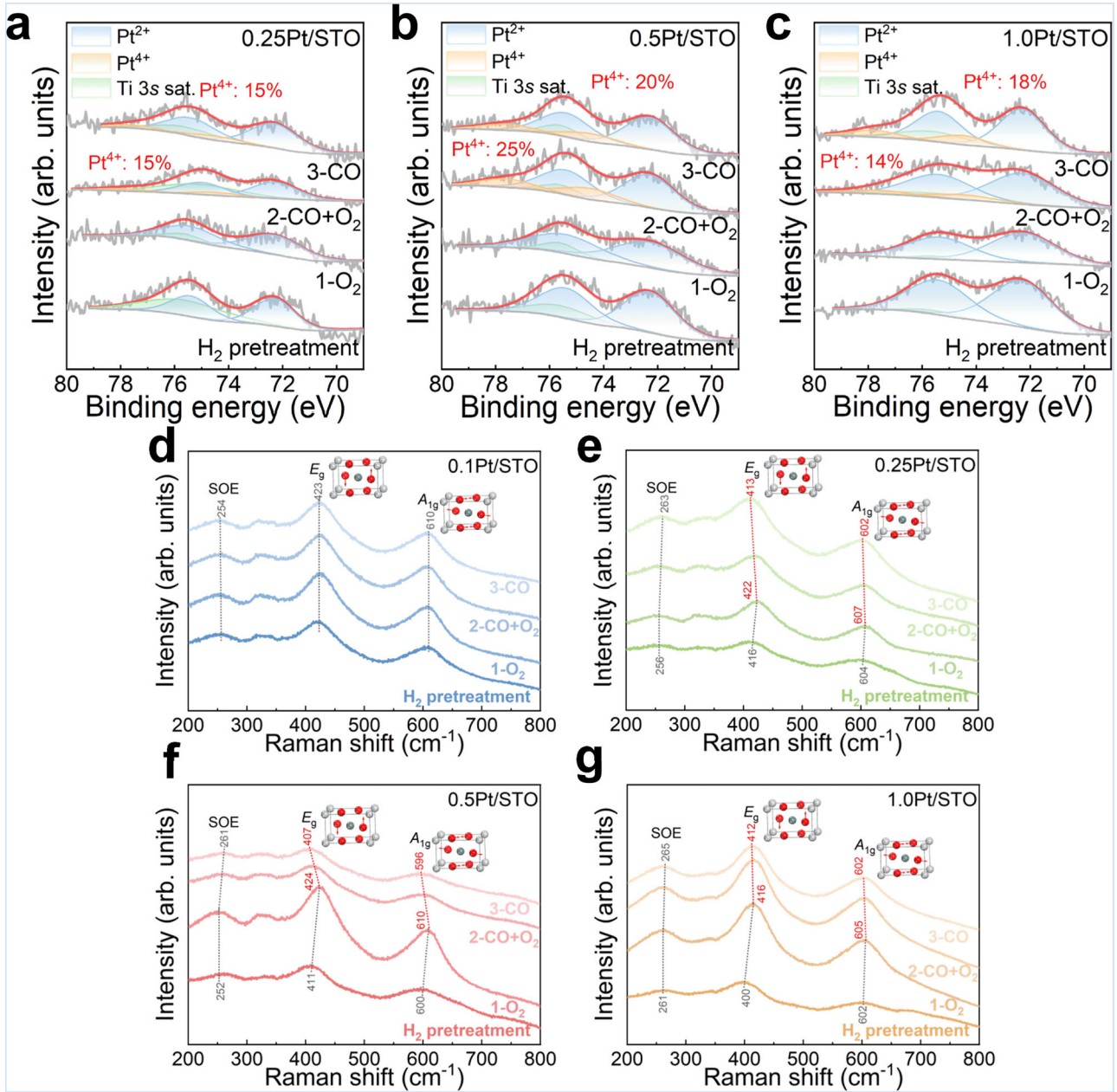

**Fig. 2 | In situ NAP-XPS and Raman spectra of Pt/STO catalysts under different gas inflows at 100 °C.** In situ NAP-XPS spectra of Pt 4$f$ for (**a**) 0.25Pt/STO, **b** 0.5Pt/STO, and **c** 1.0Pt/STO under sequential inflows of $O_2$ (1 mbar), CO (0.5 mbar) + $O_2$ (0.5 mbar), and CO (1 mbar). In situ Raman spectra of (**d**) 0.1Pt/STO, **e** 0.25Pt/STO, **f** 0.5Pt/STO, and **g** 1.0Pt/STO under sequential inflows of $O_2$ (1%), CO (1%) + $O_2$ (1%), and CO (1%).

shifted to higher wavenumbers upon exposure to $O_2$, and then shifted back to lower wavenumbers when CO + $O_2$ was introduced, with the backward shift being more pronounced when only CO was present. The downward shift of the $A_{1g}$ and $E_g$ vibrations indicates weakened O-Ti(Sn)-O bond strengths, suggesting a higher mobility of lattice oxygen ($O_{latt}$)[38], which is the prerequisite of the occurrence of ROS during the CO oxidation process. In comparison to 0.25Pt/STO and 1.0Pt/STO catalyst, the Raman peak shifts in 0.5Pt/STO were more pronounced during CO exposure. This result indirectly validates a more substantial intensity of ROS effect in the CO oxidation process for 0.5Pt/STO, suggesting that Pt in the form of nanoclusters is more effective in generating the ROS effect. In contrast, for 0.1Pt/STO catalyst, no changes in the Raman peak positions were detected during the sequential exposure to $O_2$, CO + $O_2$, and CO, indicating that the bond strength of the STO support remained unchanged during the CO oxidation reaction. This implies that 0.1Pt/STO did not manifested a

significant ROS effect during CO oxidation. Consequently, it can be inferred that there is no notable ROS effect occurring at the single-atom Pt and STO interface.

The influence of temperature on ROS was investigated using in situ NAP-XPS and in situ diffuse reflectance infrared Fourier transform spectroscopy (DRIFTS) under exposure to CO + $O_2$ gas mixture. As shown in Fig. 3a–c, no discernible $Pt^{4+}$ characteristic peaks were detected for 0.25Pt/STO and 1.0Pt/STO within the 30–50 °C range. Distinct $Pt^{4+}$ peaks only appeared when the temperature was elevated to 100 °C. In the case of 0.5Pt/STO catalyst, $Pt^{4+}$ peaks were observable at 50 °C, and the proportion of $Pt^{4+}$ increased further as the reaction temperature reached to 100 °C. These results imply a clear positive correlation between the intensity of ROS and the reaction temperature. Notably, the 0.5Pt/STO catalyst demonstrated significantly higher ROS intensity compared to 0.25Pt/STO and 1.0Pt/STO. In situ DRIFTS data provide additional validation for this conclusion. As shown in

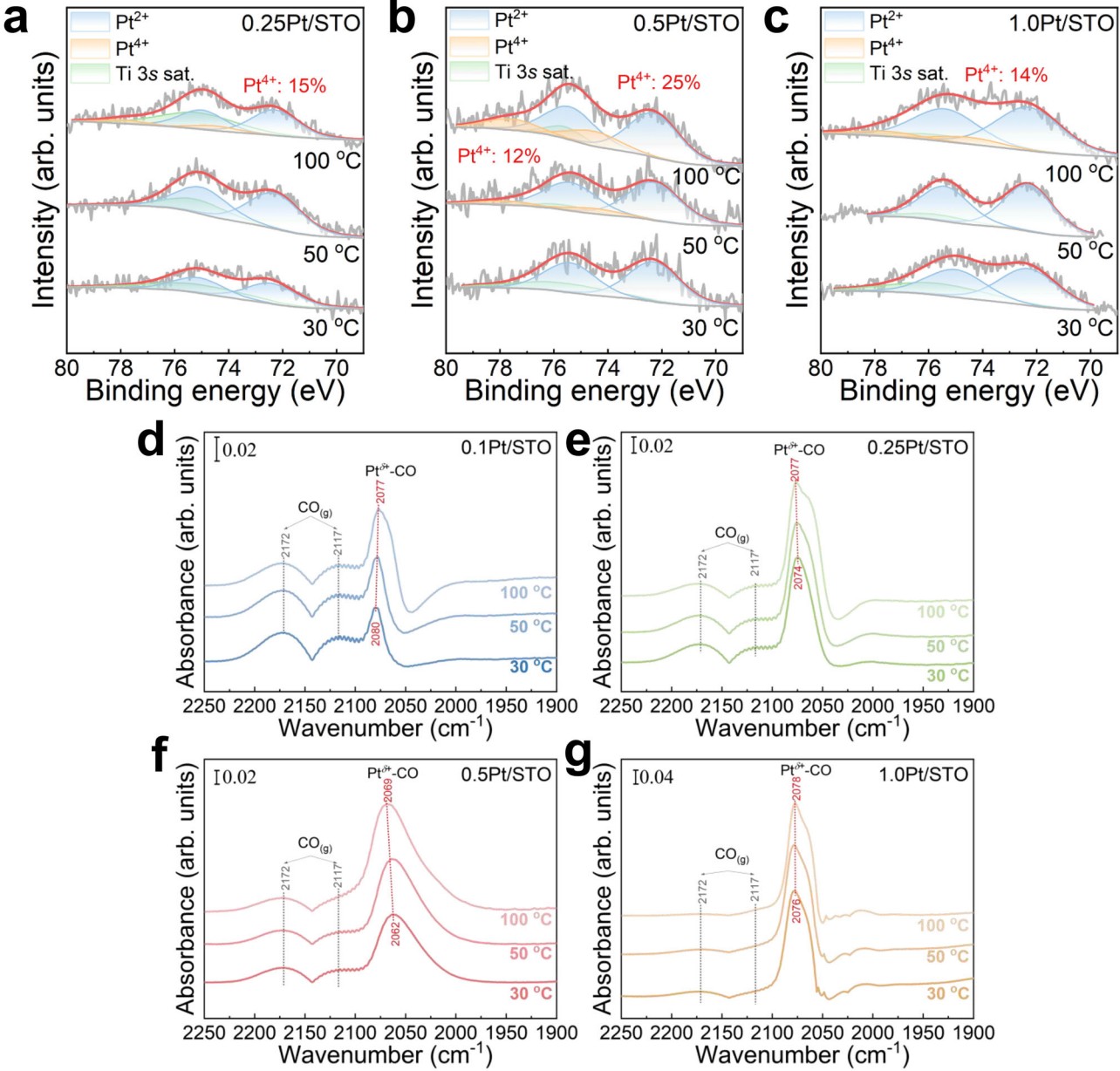

**Fig. 3 | In situ NAP-XPS and DRIFTS spectra of Pt/STO catalysts under CO flow from 30 to 100 °C.** In situ NAP-XPS spectra of Pt 4$f$ for (**a**) 0.25Pt/STO, **b** 0.5Pt/STO, and **c** 1.0Pt/STO under CO (0.5 mbar) + O$_2$ (0.5 mbar) flow at temperatures ranging from 30 to 100 °C. In situ DRIFTS spectra of (**d**) 0.1Pt/STO, **e** 0.25Pt/STO, **f** 0.5Pt/STO, and **g** 1.0Pt/STO under CO (1%) + O$_2$ (1%) flow at temperatures ranging from 30 to 100 °C.

Fig. 3d–g, all Pt/STO catalysts exhibited characteristic peaks at 2172, 2117, and ~2062 cm$^{-1}$. The peaks at 2172 and 2117 cm$^{-1}$ correspond to gaseous or weakly adsorbed CO, while the peak at ~2062 cm$^{-1}$ represents CO adsorbed on semi-oxidized Pt sites (Pt$^{\delta+}$-CO)[24,39–41]. The position of this peak is substantially influenced by the valence state of Pt[14]. The Pt$^{\delta+}$-CO peak positions for 0.1Pt/STO, 0.25Pt/STO, 0.5Pt/STO, and 1.0Pt/STO were 2080, 2074, 2062, and 2076 cm$^{-1}$, respectively, indicating that the Pt valence state on the surface of the Pt/STO catalysts varies with different Pt loadings. Interestingly, as the reaction temperature increased, the Pt$^{\delta+}$-CO peak for 0.1Pt/STO shifted towards lower wavenumbers, while those for 0.25Pt/STO, 0.5Pt/STO, and 1.0Pt/STO shifted towards higher wavenumbers. This trend indicates that the changes in the valence state of Pt during CO oxidation for 0.25Pt/STO, 0.5Pt/STO and 1.0Pt/STO are opposite to those for 0.1Pt/STO. Therefore, it can be inferred that 0.1Pt/STO does not exhibit a significant ROS effect, further supporting the notion that single-atom Pt is not conducive to ROS. Additionally, the most pronounced high-

wavenumber shift observed on the 0.5Pt/STO catalyst further suggests that Pt in the form of nanoclusters is most conducive for generating ROS effects.

**Effect of ROS on CO oxidation**

CO oxidation performance of all Pt/STO catalysts was tested to explore the relationship between ROS effects and CO oxidation activity. The Mears criterion (see Supplementary Note 2 and Supplementary Fig. S8) was applied to ensure that the activity tests were not limited by mass or heat transfer. Supplementary Fig. S9 depicts the CO oxidation conversion as a function of temperature for the Pt/STO catalysts, with the results indicating that 1.0Pt/STO exhibited the highest activity, albeit only marginally surpassing that of 0.5Pt/STO. Both 0.5Pt/STO and 1.0Pt/STO catalysts met the U.S. Department of Energy's criterion of "90% conversion of all criteria pollutants at 150 °C"[42]. To probe the intrinsic reaction rate of CO oxidation, the turnover frequency of CO oxidation at Pt sites (TOF$_{Pt}$) was calculated. As shown in Fig. 4a, the

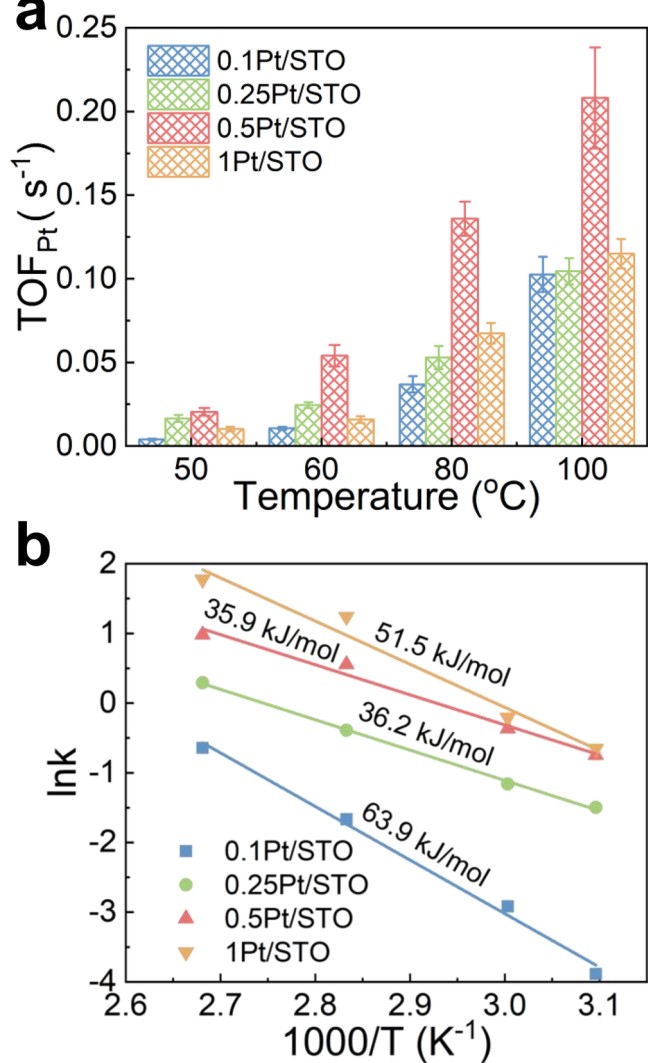

**Fig. 4 | CO oxidation performance of Pt/STO catalysts. a** CO oxidation turnover frequency at Pt sites ($TOF_{Pt}$) from 50 to 100 °C; error bars represent standard deviation. **b** Activation energy determined by the Arrhenius equation.

0.1Pt/STO catalyst, on which Pt in the form of single-atom with the highest dispersion, displayed the lowest $TOF_{Pt}$. As the Pt loading increased, Pt gradually aggregated into nanocluster (0.5Pt/STO), resulting in a $TOF_{Pt}$ 2–5 times higher than that of single-atom Pt. Further increasing Pt content led to the formation of nanocrystal (1.0Pt/STO), with a $TOF_{Pt}$ slightly higher than that of single-atom Pt. There is a remarkable parallelism between the trend of $TOF_{Pt}$ and that of ROS intensity on the Pt/STO catalysts. Specifically, single-atom Pt on 0.1Pt/STO catalyst exhibits scarcely any ROS effect and the lowest $TOF_{Pt}$ value, whereas nanocluster Pt displays the most significant ROS effect and the highest $TOF_{Pt}$, outperforming both single-atom and nanocrystal Pt on Pt/STO catalyst. This finding strongly suggests that ROS exert a substantial influence on the CO oxidation activity of Pt/STO catalysts.

Figure 4b illustrates that the trend of activation energy (Ea) for the Pt/STO catalysts is also in accordance with the variations in $TOF_{Pt}$ and ROS intensity. The activation energy for nanocluster Pt (represented by the 0.5Pt/STO catalyst, 35.9 kJ/mol) is significantly lower than that of single-atom Pt (0.1Pt/STO, 63.9 kJ/mol). Interestingly, the results in Supplementary Fig. S10 indicate that, for all Pt/STO catalysts, the reaction orders of CO and $O_2$ remain approximately constant and are slightly greater than zero. This suggests that the CO oxidation on all Pt/

STO catalysts follows the MvK mechanism[39], regardless of whether a significant ROS effect exists. Therefore, the ROS in CO oxidation introduces an additional step of oxygen migration from the support to the active sites, without altering the fundamental nature of the MvK mechanism that relies on lattice oxygen to oxidize CO.

To evaluate the practical applicability of Pt/STO catalysts, the 0.5Pt/STO sample was selected to activity tests under different gas hourly space velocities (GHSV), multiple reaction cycles, and a 24 h stability. As shown in Supplementary Fig. S11, 0.5Pt/STO exhibits excellent tolerance to high space velocities. Its activity remained nearly unchanged when the GHSV increased from 30,000 to 120,000 mL $g_{cat}^{-1}$ $h^{-1}$, and only a slight decline was observed at 240,000 mL $g_{cat}^{-1}$ $h^{-1}$. In addition, the catalytic activity was well maintained during four reaction cycles and in the 24 h stability test, indicating that Pt/STO possesses high stability for CO oxidation and strong potential for practical application.

### The essence of ROS

To explore the impact of different Pt particle sizes on ROS, ab initio molecular dynamics (AIMD) and density functional theory (DFT) simulations were conducted on single-atom, nanocluster, and nanocrystal Pt loaded (110) facet of STO support (denoted as $Pt_{sin}$/STO, $Pt_{clu}$/STO, and $Pt_{cry}$/STO, respectively). Detailed simulation setups and model construction can be found in Supplementary Notes 3-4, Supplementary Figs. S12–S14, and Supplementary Table S3. After obtaining the structural models, a CO molecule was adsorbed on the Pt sites, followed by 20 picoseconds (ps) of AIMD simulations to analyze the microscopic reaction processes of CO oxidation. The results revealed distinctly different reaction pathway for $Pt_{sin}$/STO, $Pt_{clu}$/STO, and $Pt_{cry}$/STO: For $Pt_{sin}$/STO (Fig. 5a), the structure remained highly stable after CO adsorption, showing no changes even after 20 ps of AIMD simulations; For $Pt_{clu}$/STO (Fig. 5b), the structure remained stable for the first 7 ps but then exhibited breaking of Ti-O and Sn-O bonds, while the Pt-O bond length remained unchanged, which suggests oxygen migration from the support to the active Pt site (i.e. ROS process); For $Pt_{cry}$/STO (Fig. 5c), it displayed significant instability immediately upon CO adsorption, with Pt-O bond cleavage occurring at the start of the AIMD simulation, while Ti-O and Sn-O bonds remained intact. This indicates that Pt atoms in $Pt_{cry}$/STO quickly separate from the support after CO adsorption, and subsequent CO oxidation likely involves oxygen from PtO rather than from the support.

The CO adsorption energies of the initial structures in Fig. 5a–c were calculated using DFT optimization (Supplementary Fig. S15). $Pt_{sin}$/STO exhibited a significantly higher adsorption energy (−2.77 eV) compared to $Pt_{clu}$/STO (−0.84 eV) and $Pt_{cry}$/STO (−1.20 eV), suggesting that the stability of CO adsorption on $Pt_{sin}$/STO is likely the primary reason for its structural stability during AIMD simulations. Furthermore, the optimized structure of CO adsorbed on $Pt_{cry}$/STO (Supplementary Fig. S15) matched the final structure from AIMD simulations (Fig. 5c), confirming that CO adsorption at the Pt interface causes Pt detachment from the interface. Supplementary Fig. S16 provides the structures and adsorption energies of $O_2$ adsorbed on $Pt_{clu}$/STO, showing negligible adsorption. Hence, CO oxidation on $Pt_{clu}$/STO likely follows the MvK mechanism, consistent with kinetic studies.

To further investigate the impact of CO adsorption induced charge transfer on the microscopic structure, charge density differences and Bader charge changes before and after CO adsorption were calculated to quantify the extent of charge transfer (see Fig. 5d–f). The left panels of Fig. 5d–f indicate that CO adsorption on $Pt_{clu}$/STO results in significant charge accumulation near interfacial oxygen. This accumulation weakens the bonding requirements of the interfacial oxygen, facilitating its migration to the active site (i.e. ROS process). A similar phenomenon is observed when CO is adsorbed on $Pt_{sin}$/STO, but the amount of charge accumulation is markedly lower and insufficient to support the ROS effect, thereby maintaining

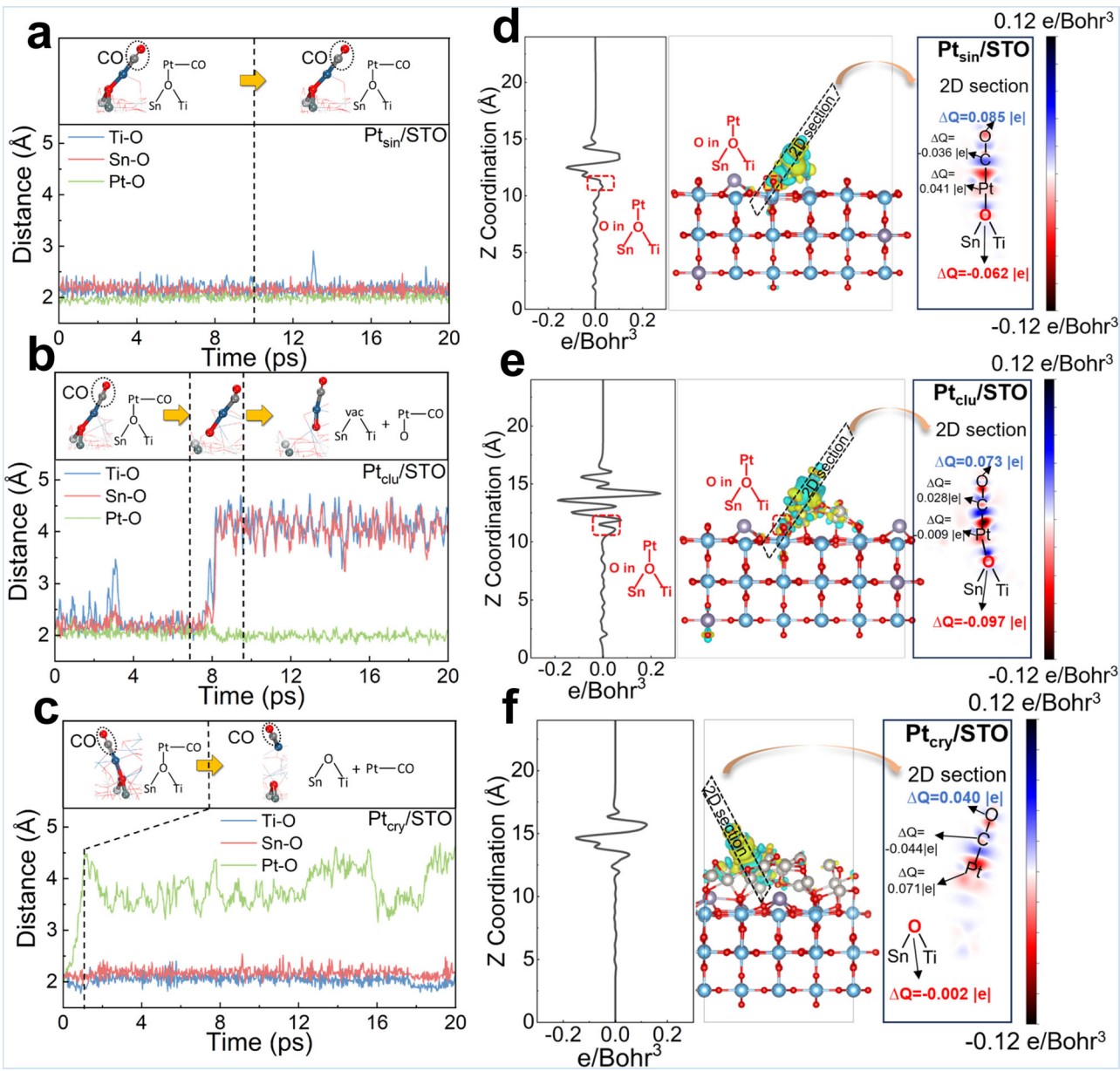

**Fig. 5 | AIMD simulations and charge density differences of CO adsorption on Pt/STO catalysts with different Pt sizes.** Interatomic distances of Pt-O, Ti-O, and Sn-O during AIMD simulations of CO adsorption on (**a**) $Pt_{sin}$/STO, **b** $Pt_{clu}$/STO, and **c** $Pt_{cry}$/STO. Charge density difference analysis after CO adsorption on (**d**) $Pt_{sin}$/ STO, **e** $Pt_{clu}$/STO, and **f** $Pt_{cry}$/STO; each composite image includes the following from left to right: plane-averaged charge density changes along the z-direction, 3D charge density difference (yellow represents increased charge density, blue represents decreased charge density), and 2D section of charge density differences.

structural stability during AIMD simulations. In contrast, CO adsorption on $Pt_{cry}$/STO leads to the detachment of the adsorption site Pt atom from the interface, which minimizes the influence of charge transfer on the interfacial oxygen. The right panels of Fig. 5d–f provide a quantitative analysis of the Bader charge changes for each atom after CO adsorption, with detailed data presented in Supplementary Table S4. It is important to note that the Bader charge values do not represent absolute oxidation states but instead reflect trends in oxidation state changes. When CO is adsorbed on $Pt_{sin}$/STO, the valence state of interfacial oxygen decreases by 0.062 | e |, while on $Pt_{clu}$/STO, it decreases by 0.097 |e|. This demonstrates that CO adsorption on $Pt_{clu}$/STO leads to greater electron accumulation at the interfacial oxygen, promoting the ROS effect. The source of these electrons enriched at the interfacial oxygen primarily originates from the oxygen atom in CO, whose valence state increases by approximately 0.073−0.085 |e| after adsorption.

Conversely, for CO adsorption on $Pt_{cry}$/STO, the valence state of the interfacial oxygen remains essentially unchanged. In this case, electronic rearrangement predominantly occurs within the CO molecule, where the valence state of carbon decreases by 0.044 |e| and that of oxygen increases by 0.040 |e|.

## Effect of ROS on reaction cycle
To investigate the effect of different Pt particle sizes on the CO oxidation reaction, the reaction cycles for CO oxidation on $Pt_{sin}$/ STO, $Pt_{clu}$/STO, and $Pt_{cry}$/STO were calculated using DFT based on AIMD simulation results. These included CO oxidation by $O_{latt}$ from the support on the $Pt_{sin}$/STO surface, CO oxidation by ROS on the $Pt_{clu}$/STO surface, and CO oxidation by $O_{latt}$ from PtO on the $Pt_{cry}$/ STO surface (see Fig. 6a−c). The CO oxidation pathways on the $Pt_{sin}$/ STO and $Pt_{cry}$/STO surfaces (configurations i−vii in Fig. 6a, c) were almost identical. In both cases, CO adsorbs onto the Pt site (i→ii) and

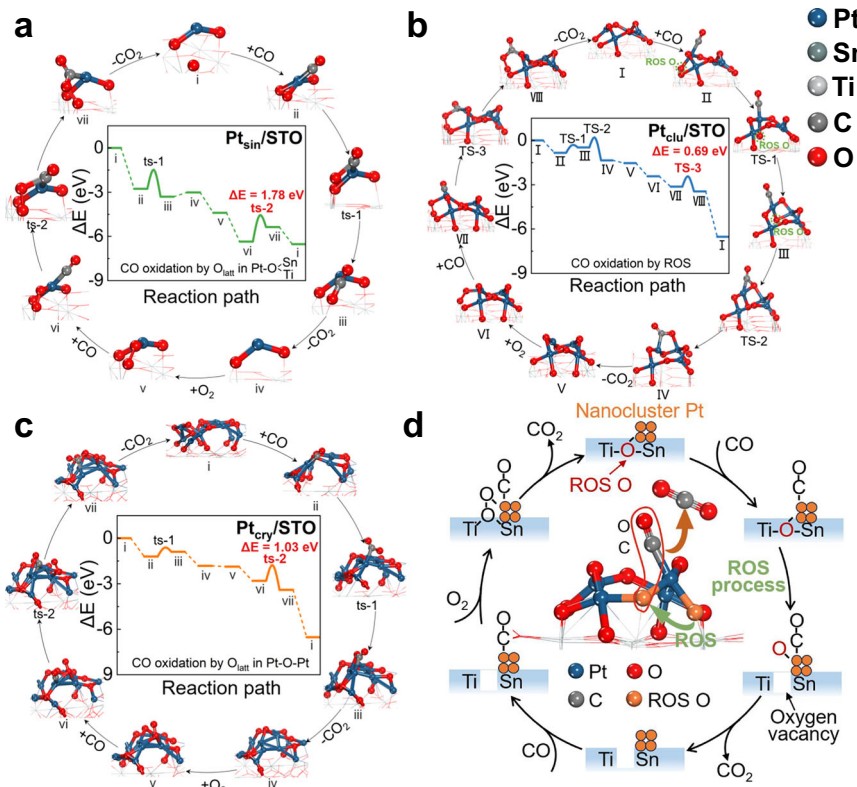

**Fig. 6 | CO oxidation reaction cycles on Pt/STO catalysts with different Pt sizes.** CO oxidation reaction cycles and energy profiles for (**a**) $Pt_{sin}$/STO, **b** $Pt_{clu}$/STO, and **c** $Pt_{cry}$/STO. **d** Schematic illustration of the CO oxidation mechanism on PtO clusters via a reverse oxygen spillover pathway to form $CO_2$ on the Pt/STO catalyst surface.

is subsequently oxidized to $CO_2$ by $O_{latt}$—either from the support ($Pt_{sin}$/STO) or PtO ($Pt_{cry}$/STO) (ii→ts-1→iii). After the desorption of the generated $CO_2$, an oxygen vacancy ($O_v$) is formed (iii→iv), which is subsequently filled by $O_2$ adsorption (iv→v). A second CO molecule then adsorbs (v→vi) and is oxidized to $CO_2$ by the adsorbed $O_2$ (vi→ts-2→vii). Upon the desorption of the newly formed $CO_2$, the catalyst returns to its initial state, completing the reaction cycle (vii→i). In this cycle, the rate-limiting step for both $Pt_{sin}$/STO and $Pt_{cry}$/STO is the oxidation of CO by the adsorbed $O_2$ (vi→ts-2→vii). However, the reaction energy barrier for $Pt_{sin}$/STO (1.78 eV) is significantly higher than that for $Pt_{cry}$/STO (1.03 eV). This difference likely arises from the excessively strong CO adsorption energy on $Pt_{sin}$/STO (−2.77 eV), which inhibits the oxidation of adsorbed CO.

The CO oxidation pathway on $Pt_{clu}$/STO is also similar to those on $Pt_{sin}$/STO and $Pt_{cry}$/STO, with the addition of a ROS step. As shown in Fig. 6b, CO adsorbs onto a Pt site on $Pt_{clu}$/STO (I → II), triggering the migration of lattice oxygen ($O_{latt}$) from the support to the Pt site (i.e., ROS, II → TS-1 → III), leaving an $O_v$ on the support. The adsorbed CO is then oxidized to $CO_2$ by the $O_{latt}$ within $Pt_{clu}$, and the consumed O is replenished by the ROS-derived O (III → TS-2 → IV). After the $CO_2$ desorbs (IV → V), $O_2$ adsorbs at the $O_v$ on the support (V → VI), and another CO molecule adsorbs onto the Pt site (VI → VII). The adsorbed CO is then oxidized by $O_2$, completing the cycle with the desorption of $CO_2$ and the restoration of the catalyst to its initial state (VII → TS-3 → VIII → I). This reaction cycle, facilitated by the ROS effect, allows CO to be oxidized by $O_{latt}$ of $Pt_{clu}$ while leaving an $O_v$ on the support. It leverages the high oxidation capacity of the active site and the strong $O_2$ adsorption capacity of the support, enhancing the redox reaction cycle. The energy barrier for this reaction cycle (0.69 eV) corresponds to the oxidation of CO by the adsorbed $O_2$ (VII → TS-3 → VIII), which is significantly lower than those for $Pt_{sin}$/STO (1.78 eV) and $Pt_{cry}$/STO (1.03 eV).

## Discussion

In our previous study[19], we doped Sn atoms into the $TiO_2$ support (referred to as the STO support) to create asymmetric oxygen sites and facilitate the migration of lattice oxygen of the support. Consequently, we established the interfacial process of ROS on Pt/STO catalyst, in which CO adsorption is the prerequisite. This ROS effect significantly enhanced the CO oxidation activity of the Pt/STO catalyst. However, the mechanism through which different forms of active Pt sites influence the ROS effect remains a critical topic for further investigation. To address this issue, in the present work we synthesized a series of Pt/STO catalysts with varying Pt loadings, ranging from single-atom Pt (0.1Pt/STO) to nanocluster (0.5Pt/STO) and nanocrystal (1.0Pt/STO). Transient CO oxidation reactions conducted under 1% $CO/N_2$ revealed that nanocluster Pt exhibited significantly more active oxygen species available for CO oxidation than single-atom or nanocrystal Pt. Notably, these active oxygen species could not be characterized via $O_2$-TPD, suggesting that nanocluster Pt possess a higher amount of "in situ-generated oxygen".

A series of in situ characterizations were performed to investigate the source of this "in situ-generated oxygen". Results from in situ NAP-XPS and in situ Raman spectra showed that the active Pt sites and the STO support in Pt/STO catalysts remain stable under oxidizing ($O_2$) conditions. However, the Pt sites in $Pt_{clu}$/STO and $Pt_{cry}$/STO were oxidized under reducing (CO) conditions, accompanied by oxygen migration from the STO support (i.e. ROS effect). Notably, the ROS effect was more pronounced in $Pt_{clu}$/STO compared to $Pt_{cry}$/STO. In contrast, $Pt_{sin}$/STO remained stable under reducing (CO) conditions, indicating no occurrence of the ROS effect. In situ NAP-XPS and in situ DRIFTS spectra at varying reaction temperatures further confirmed these observations, revealing that the ROS effect intensifies with increasing temperature. When compared with catalysts lacking the ROS effect, such as 0.1Pt/STO in this study, catalysts with a more prominent ROS effect (e.g., 0.5Pt/STO) exhibited a $TOF_{Pt}$ 2–5 times

higher. This finding holds great significance for the design of noble-metal catalysts.

Although CO chemisorption constitutes an essential precondition for ROS effect, the substantial interfacial chemistry of CO adsorption is closely related to the size of Pt nanoparticles on $Pt/TiO_2$ catalysts. The microscopic mechanisms underlying the influence of Pt particle size on the ROS effect and its promotion of CO oxidation were investigated using charge density difference analysis, AIMD, and reaction cycle simulations. The $Pt_{clu}/STO$ configuration exhibited the strongest ROS effect, which was primarily attributed to strong electrons transfer to the interfacial oxygen species ($\triangle Q = -0.097\ |e|$), thereby enhancing their migration potential. A comparable phenomenon was observed in $Pt_{sin}/STO$, but the quantity of charge transferred to the interfacial oxygen was significantly lower ($\triangle Q = -0.062\ |e|$). Moreover, the significantly more negative CO adsorption energy for $Pt_{sin}/STO$ (−2.77 eV vs. −0.84 eV for $Pt_{clu}/STO$ and −1.20 eV for $Pt_{cry}/STO$) indicates the thermodynamical stabilization of CO-adsorbed state, thereby kinetically hindering subsequent ROS-mediated oxidation steps. In contrast, for $Pt_{cry}/STO$, CO adsorption caused a pronounced decoupling between adsorption sites and oxygen reservoirs (the support), which creating a kinetic bottleneck for ROS effect. Reaction cycle simulations further confirmed that $Pt_{clu}/STO$ exhibits the lowest activation barrier (0.69 eV) for the rate-limiting CO oxidation step, in excellent agreement with experimental catalytic activity evaluation.

Based on mechanistic insights, we propose a modified Mars-van Krevelen (MvK) pathway incorporating ROS-mediated oxygen mobilization (Fig. 6d): (i) CO chemisorption-induced oxygen activation at metal-support interfaces, (ii) surface migration of activated oxygen species to metal sites (ROS process), and (iii) Langmuir-Hinshelwood type $CO_2$ formation. This refined mechanism introduces an additional ROS step prior to the conventional MvK redox cycle which is closely related to the size of Pt nanoparticles, representing a distinct interfacial catalysis paradigm. We believe that ROS is not an incidental phenomenon confined to a single system but rather a general process that should also occur in systems with similar properties. Therefore, a systematic investigation is needed to examine the effects of different dopants−including transition metals and nonmetals−on ROS. Moreover, the role of different supports (such as $CeO_2$ and $Al_2O_3$) should also be explored. These studies will not only deepen the understanding of the fundamental factors governing ROS, but also offer valuable guidelines for the rational design of advanced catalysts with enhanced performance.

## Methods
### Synthesis of materials
The $Sn_{0.2}Ti_{0.8}O_2$ (STO) support was prepared by co-precipitation method. The molar ratio of 2: 8 of $SnCl_4·5H_2O$ and $Ti(SO_4)_2$ were dissolved in deionized water. Ammonia solution (25 wt%) was then added to the solution until the pH reached 10, prompting the co-precipitation of Sn and Ti ions, which were filtered and washed with deionized water until neutralized. The resulting materials were dried at 105 °C and calcined at 500 °C for 4 h in air. The heating rate was 2 °C/min.

All the Pt/STO catalysts were synthesized by impregnation method. The platinum precursor was $Pt(NO_3)_2$ solution. Suitable amounts of $Pt(NO_3)_2$ (corresponding to 0.1, 0.25, 0.5 and 1.0 wt% Pt) were added into 2 g STO support and then diluted to 10 mL with vigorous stirring. The obtained samples were dried at 105 °C and calcined at 500 °C under air for 1 h under the heating rate of 2 °C/min. After calcination, the catalysts were reduced in 5% $H_2/N_2$ at 300 °C for 1 h. The primary purpose of this treatment was to lower the valence state of surface Pt, thereby facilitating the ROS effect and enhancing the CO oxidation activity of the catalyst[19]. Since the precursors used for catalyst synthesis contained S and Cl elements, their possible residues could affect the catalytic performance. Therefore, XPS was employed to examine the residual S and Cl species of all Pt/STO catalysts. As

shown in Supplementary Fig. S17, no S or Cl signals were detected on the surfaces of any Pt/STO catalysts, indicating that the influence of residual S and Cl on the catalysts can be considered negligible.

### CO oxidation performance
The catalytic performance in CO oxidation was evaluated by a fixed-bed quartz micro-reactor with 100 mg of catalyst (40−60 mesh). The typical reaction conditions involved 1% CO, 1% $O_2$, and $N_2$ as the balance gas. The total flow rate was 100 mL/min, yielding a gas hourly space velocity (GHSV) of 60,000 mL $g_{cat}^{-1}$ $h^{-1}$. CO and $CO_2$ concentrations in both the inlet and outlet streams were measured using infrared gas analyzer (Gasmet Dx-4000). CO conversion was determined by the following formula:

$$\eta_{CO} = \frac{C_{CO_{in}} - C_{CO_{out}}}{C_{CO_{in}}} \qquad (1)$$

where, $\eta_{CO}$ (%), $C_{COin}$ (µmol s$^{-1}$) and $C_{COout}$ (µmol s$^{-1}$) represent CO conversion rate, CO concentrations in the inlet and outlet, respectively.

The CO oxidation reaction rate $k$ (µmol g$^{-1}$ s$^{-1}$) and TOF (s$^{-1}$) were measured under conditions that maintained CO conversion below 20% to minimize diffusion limitations.

$k$ was calculated assuming ideal gas behavior:

$$k = \frac{\eta_{CO} \cdot C_{CO_{in}}}{W} \qquad (2)$$

where, $k$ (µmol g$^{-1}$ s$^{-1}$), $\eta_{CO}$ (%), $C_{COin}$ (µmol s$^{-1}$), and $W$ (g) represent the reaction rate, CO conversion rate, CO concentrations in the inlet and catalyst mass.

TOF also calculated assuming ideal gas behavior by the following equation:

$$TOF_{Pt} = \frac{\eta_{CO} \cdot C_{CO_{in}}}{N_{surfPt}} \qquad (3)$$

where, $TOF_{Pt}$ (s$^{-1}$), $\eta_{CO}$ (%), $C_{COin}$ (µmol s$^{-1}$), and $N_{surfPt}$ (µmol) represent CO oxidation turnover frequency over Pt sites, CO conversion rate, CO concentrations in the inlet and the number of Pt sites, respectively.

The apparent activation energies ($E_a$, kJ mol$^{-1}$) were derived from the Arrhenius equation, using $k$ obtained when CO conversion rates below 20%.

### Characterization
XRD patterns were acquired using a D8 Advance X-ray diffractometer (Bruker AXS) with Cu K$_\alpha$ radiation. The scan was performed over a 2θ range of 5−90° at a scan rate of 4°/min and a step size of 0.02°. The instrument operated at a voltage of 40 kV and a current of 40 mA.

$N_2$ physisorption measurements were conducted at liquid nitrogen temperature using a Micromeritics ASAP 2460 in static mode. Prior to analysis, the catalysts were degassed at 300 °C for 4 h. The specific surface area was calculated using the Brunauer-Emmett-Teller (BET) method, while pore volume and average pore diameter were determined via the Barrett-Joyner-Halenda (BJH) method from the $N_2$ adsorption-desorption isotherms.

$O_2$-TPD experiments were performed on a Micromeritics Auto-Chem 2920 ThermoStar chemisorption analyzer. The catalyst was pretreated in a 2% $O_2$/He flow for 1 h at 300 °C, then cooled to 50 °C for 30 min in the same gas mixture. Weakly adsorbed O species were purged with He for 30 min before the temperature was raised to 800 °C at a rate of 10 °C min$^{-1}$ in a He flow. $O_2$ desorption was monitored by thermal conductivity detector (TCD).

In situ NAP-XPS spectra were recorded with a SPECS-AU190069 system equipped with a static voltage lens and multi-stage differential

pumping system, enabling ultra-high vacuum ($1 \times 10^{-9}$ mbar) or gas pressures up to 5 mbar. Monochromatized Al $K_\alpha$ radiation (1486.6 eV) was used for excitation at 50 W power. The X-ray spot diameter was approximately 0.3 mm. The reaction pressure was maintained at 1 mbar, and the sample was heated during measurements using a specially designed sample holder. An electron flood gun compensated for any charge accumulation on the catalyst.

In situ DRIFTS measurements were performed on a Nicolet iS50 FTIR spectrometer with an in situ cell and an MCT detector. All the spectra were recorded by accumulating 64 scans with a resolution of 4 cm$^{-1}$. The catalyst was pretreated at 300 °C in 5% H$_2$/He for 1 h, followed by He purging for 1 h to remove moisture and impurities. The catalyst was then cooled to the desired temperature, and spectra were recorded after introducing 1% CO and/or 1% O$_2$ in N$_2$ balance for 30 min.

In situ Raman spectroscopy was carried out using a Horiba LabRAM HR Evolution instrument with a 532 nm laser source, Synapse CCD detector, and Linkam CCR1000 in situ reactor. The catalyst was pretreated at 300 °C in 5% H$_2$/N$_2$ for 1 h and then purged with N$_2$. After cooling to 100 °C, Raman spectra were collected by averaging 8 scans with 6 s acquisition times. The reaction was initiated by introducing 1% CO and/or 1% O$_2$ in N$_2$ balance, and spectra were recorded after 30 min.

XANES and EXAFS analyses were conducted at the Singapore Synchrotron Light Sources. The radiation was monochromatized with a Si (111) double crystal monochromator, and the data were processed using Athena software.

AC-STEM imaging was performed using a JEM ARM200F transmission electron microscope operating at 200 kV, equipped with EDX detector, high-angle annular dark-field detector and probe corrector.

DFT and AIMD simulation were performed using Vienna Ab initio Simulation Package (VASP 5.4.4). Detailed simulation setups and model construction can be found in Supplementary Notes 3-4, Supplementary Figs. S12–S14, and Supplementary Table S3.

## Data availability
The Figs. 1–5 data generated in this study are provided in the Source Data file. Data are available from the corresponding authors upon request. Source data are provided with this paper.

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

## Acknowledgements
This research was supported by National Key R&D program of China (No. 2022YFC3701600 and 2023YFC3710700 to J.J.C.), National Natural Science Foundation of China (No. 22206155 and 22576168 to S.C.X.), China Postdoctoral Science Foundation (No. 2022M712632 and 2023T160547 to S.C.X.), Sichuan Science and Technology Program (No. 2025ZNSFSC1257 to H.L.W.), Fundamental Research Funds for the Central Universities of China (No. 2682025CX016 to H.L.W.).

## Author contributions
J.J.C. and S.C.X. proposed idea. S.C.X. wrote the manuscript and performed DFT simulation. Z.J.G. and H.L.W. modified the manuscript. J.J.C. and J.H.Li. supervised the whole work. H.Y.L., J.Q.S., J.X.M. and X.P.C. performed all the experiments. All the authors discussed the results and commented on the manuscript.

## Competing interests
The authors declare no competing interests.
