## [Transparent Peer Review file · Nature Communications]

Pt size-dependent reverse oxygen spillover on Sn-doped Pt/TiO₂ for CO oxidation

Corresponding Author: Professor Jianjun Chen

Version 0:

Reviewer comments:

Reviewer #1

(Remarks to the Author)

This study investigates the reverse oxygen spillover (ROS) phenomenon in Sn-doped Pt/TiO₂ (Pt/STO) catalysts and its dependence on Pt particle size for CO oxidation. ROS, where oxygen migrates from the support to Pt active sites, significantly influences catalytic performance. Using experimental characterizations and ab initio molecular dynamics (AIMD) simulations, the study reveals that nanocluster Pt exhibits the strongest ROS effect, leading to the highest turnover frequency for CO oxidation, approximately twice that of single-atom and nanocrystal Pt.

Overall, the manuscript is well organized and well written. I think this manuscript can be published in Nature Communications. However, several issues should be addressed before accepting the manuscript.

1. The ionic radii of Sn⁴⁺ and Ti⁴⁺ are different. Why the lattice constant of TiO₂ does not change by Sn substitution in XRD (Fig. S1)? Reasonable explanation for this is necessary.

2. The reviewer could not understand the justification that "The aberration-corrected high-angle annular dark-field scanning transmission electron microscopy results (Supplementary Figs. S3 and S4) demonstrate that both crystalline Pt and the STO support predominantly expose the (110) crystal plane" at line 105, Page 5. Since TEM images are projection image, the fact that (110) lattice fringe is observed does not mean (110) plane is exposed. This is a very critical point since the structure model used for theoretical calculations are based on this assumption.

3. Although the bright spots in the HAADF images are assigned to Sn, and Pt, this interpretation should be approached with caution. Given that atoms are aligned in the depth direction and their positions can affect the image contrast, it is not possible to reliably assign elements based solely on brightness in HAADF images.

4. The reviewer could not understand the structural model of Pt_{cryst}/STO. It appears that a PtO layer is deposited on the STO surface. Is this interpretation correct? Additionally, the reaction seems to proceed exclusively on the PtO layer in a L-H manner. If so, is it reasonable to neglect the potential effect of the Pt-STO interface or the periphery sites?

Reviewer #2

(Remarks to the Author)

This manuscript reports a study on Pt size-dependent reverse oxygen spillover (ROS) behavior in CO oxidation over Sn-doped rutile TiO₂ (STO) supports. By combining in situ characterizations with AIMD and DFT simulations, the authors claim to uncover a size-specific trend in ROS intensity, suggesting nanocluster Pt (~0.5 wt%) shows the most pronounced ROS effect and the highest intrinsic activity. While the mechanistic framework is conceptually clear and the methodology is generally sound, the novelty of the study is somewhat limited due to its incremental nature relative to the authors' own prior work (Nat. Commun. 2022), which already established the ROS phenomenon in Pt/STO systems. Several issues need to be clarified or improved, particularly in terms of justification of material choice, the true novelty of the size-dependence finding, the applicability of the reaction conditions, and the speculative nature of key mechanistic claims (e.g., in situ generation of active oxygen). Below are detailed comments:

(1) The authors should clarify the purpose of the H₂ pre-treatment step. Was it intended to remove surface oxygen species, adjust the Pt oxidation state to Pt²⁺, or promote strong metal-support interactions? What was the direct evidence of its effect,

and how does it relate to the proposed ROS mechanism?

(2) The manuscript would benefit from a more detailed justification for using Sn-doped rutile TiO₂. While the previous study introduced the Sn-induced asymmetric Ti-O-Sn sites, it remains unclear whether rutile TiO₂ is chosen for its inherent properties (e.g., stability, crystal facet exposure) or solely for consistency. Were other TiO₂ polymorphs or doping elements considered?

(3) Given the authors' prior work in Nat. Commun. already demonstrated the ROS effect on Pt/STO systems, this manuscript appears to focus mainly on Pt size effects, an expected variable in catalysis. The authors need to better emphasize what fundamentally new insight this study provides beyond confirming that nanoclusters show better performance (a commonly observed trend in Pt-based catalysis for CO oxidation).

(4) The transition from single atoms to nanocrystals with increasing Pt loading is quite expected. The corresponding decrease in Pt dispersion and its impact on CO oxidation activity is also well-documented. The authors should be cautious not to overstate this as a novel finding and should instead focus on the mechanistic details of ROS involvement that may be size-specific.

(5) Since Ti(SO₄)₂ and SnCl₄ were used as precursors, residual S or Cl species could be present on the STO surface, potentially affecting Pt dispersion, oxidation state, or ROS. The authors should clarify whether these elements were detected (via XPS, ICP-OES, etc.) and whether any such residues impact the observed catalytic behavior.

(6) Line 156 speculates that in situ-generated active oxygen species cannot be detected by O₂-TPD. However, this lacks direct experimental validation. Isotopic labeling experiments (e.g., ¹⁸O₂ vs. ¹⁶O₂) would be highly informative to confirm oxygen migration pathways. The claim of in situ-generated ROS would be far more compelling with such evidence.

(7) The CO oxidation tests were performed at relatively low gas hourly space velocities (GHSV), which are far from realistic exhaust conditions. Furthermore, only one cycle was shown per catalyst, with no assessment of durability or possible structural evolution (e.g., Pt sintering or migration). This limits the practical relevance of the findings.

(8) Realistic exhaust or flue gas streams contain water vapor and often sulfur-containing species. The study lacks any examination of catalyst performance under such conditions. What is the target application of this catalyst? If the study is primarily mechanistic, this should be made explicit, and future studies should be planned accordingly.

(9) The current findings may be specific to Sn-doped rutile TiO₂ systems. The authors should discuss whether the ROS mechanism observed here could be extrapolated to other supports or dopants. If it is a highly system-specific effect, its broader catalytic relevance is limited.

(10) Theoretical models were based on as-prepared catalysts, yet ROS and particle restructuring could occur during catalytic testing. Therefore, the models may not reflect the true stabilized catalytic surfaces. The authors should comment on this limitation and whether alternative models (e.g., post-reaction structures) were considered.

Version 1:

Reviewer comments:

Reviewer #1

(Remarks to the Author)

I think the authors have revised the manuscript satisfactorily except for the reviewer's second comment. The authors insist that (110) plane is parallel to the particle surface and/or Pt/STO interface, but the reviewer does not recognize surface nor interface in the ADF images. Where are they? The authors seem not understand the reviewer's comment that "TEM images are projection images". It is very difficult to believe that main exposed facet of the STO nanoparticles with indefinite structure is (110) at least from the AC-HAADF-STEM images shown.

Reviewer #2

(Remarks to the Author)

The authors have responded to my previous comments with a level of thoroughness and technical depth that substantially strengthens the manuscript. All major concerns raised regarding novelty, mechanistic interpretation, experimental validation, and practical relevance have been carefully addressed through new experiments, revised analyses, and clearer framing of the study's scope. In particular, the authors have convincingly clarified the conceptual advance of this work relative to their prior publications by shifting the focus from activity trends to the size-dependent modulation of reverse oxygen spillover (ROS) and its mechanistic origin. The revised discussion now articulates how CO adsorption strength, interfacial electron transfer, and Pt structural evolution jointly determine ROS intensity, rather than reiterating expected dispersion-activity relationships. This significantly improves the fundamental insight provided by the study. Several points that were previously speculative are now supported by additional experimental evidence. Notably, the inclusion of ¹⁸O isotope-labeling experiments provides direct validation of in situ lattice oxygen participation in CO oxidation, strengthening the central ROS claim beyond indirect inference from O₂-TPD alone. The added GHSV-dependent activity tests, cycling experiments, and 24 h stability data also address concerns regarding kinetic relevance and catalyst robustness, demonstrating that the conclusions are not artifacts of a narrow operating window. From a theoretical standpoint, the authors have adequately justified their structural models and clarified the treatment of interfacial effects and catalyst restructuring through AIMD simulations. While long-term catalyst aging remains beyond the scope of the present study, this limitation is now explicitly acknowledged and does not undermine the mechanistic conclusions targeted here. Overall, the revised manuscript presents a coherent and well-supported mechanistic framework for Pt size-dependent ROS behavior. In its current form, the work meets the standards of Nature Communications in rigor, clarity, and conceptual contribution. I therefore recommend acceptance without further revision.

Detailed Response to Reviewers

Jianjun Chen, Ph.D, Associate Professor

School of Environment

Tsinghua University

Beijing 100084, PR China

Email: chenjianjun@tsinghua.edu.cn

Manuscript ID: NCOMMS-25-32352

Title: Pt size-dependent reverse oxygen spillover on Sn-doped Pt/TiO₂ for CO oxidation

Dear Reviewers,

Thank you for your insightful critiques and constructive suggestions, which have significantly strengthened the scientific rigor and clarity of our manuscript. We have meticulously addressed every comment through textual revisions, data supplementation, and methodological refinements. The questions and comments were marked with a gray background; The responses to the reviewer's questions and comments were marked in blue; All the revisions and corresponding descriptions in our revised manuscript were marked in green. We confirm that this revised version now comprehensively resolves the raised concerns. Should further clarifications be needed, we stand ready to provide additional information promptly.

Yours sincerely,

Jianjun Chen

Response to Reviewer #1 (Pages 2–9)

Response to Reviewer #2 (Pages 10–24)

Response to Reviewer #1:

General comments: This study investigates the reverse oxygen spillover (ROS) phenomenon in Sn-doped Pt/TiO₂ (Pt/STO) catalysts and its dependence on Pt particle size for CO oxidation. ROS, where oxygen migrates from the support to Pt active sites, significantly influences catalytic performance. Using experimental characterizations and ab initio molecular dynamics (AIMD) simulations, the study reveals that nanocluster Pt exhibits the strongest ROS effect, leading to the highest turnover frequency for CO oxidation, approximately twice that of single-atom and nanocrystal Pt.

Overall, the manuscript is well organized and well written. I think this manuscript can be published in Nature Communications. However, several issues should be addressed before accepting the manuscript.

Response: We sincerely appreciate your valuable suggestions, which have significantly contributed to improving the innovation and readability of our manuscript. We have revised the content accordingly, and a point-by-point reply to your suggestions is provided below.

1. The ionic radii of Sn⁴⁺ and Ti⁴⁺ are different. Why the lattice constant of TiO₂ does not change by Sn substitution in XRD (Fig. S1)? Reasonable explanation for this is necessary.

Response: We fully agree that the difference in atomic radii between Sn⁴⁺ and Ti⁴⁺ can cause changes in the lattice constant of TiO₂ as reflected in XRD patterns. This phenomenon has been reported in our previous work (*Nat Commun* 2023, 14, 3477). However, in Fig. S1 we primarily compared the XRD patterns of Pt/STO catalysts with different Pt loadings. Since the peak positions and shapes remain essentially unchanged, this only indicates that varying the Pt content has a negligible effect on the crystal

structure of the STO support. We suspect that the description in the main text may have led to this misunderstanding. Therefore, we have revised the wording of the XRD discussion to minimize potential ambiguity.

Corresponding text in revised Manuscript:

Page 5:

The XRD patterns of all Pt/STO catalysts (Supplementary Fig. S1) perfectly match that of rutile TiO₂ (JCPDS: #21-1276), with no peaks attributable to PtO_x or SnO_x. The main diffraction peak at 27.2° shows a slight shift compared with the standard pattern of TiO₂ (27.3°), indicating the successful doping of Sn into the TiO₂ lattice. Moreover, the main peak position and half-peak width of all Pt/STO catalysts remain essentially unchanged, confirming that Pt loading has little influence on the crystal structure of the STO support.

Corresponding figure in revised Manuscript:

Page S7:

Supplementary Fig. S1 XRD spectra of Pt/STO catalysts.

2. The reviewer could not understand the justification that “The aberration-corrected high-angle annular dark-field scanning transmission electron microscopy results (Supplementary Figs. S3 and S4) demonstrate that both crystalline Pt and the STO support predominantly expose the (110) crystal plane” at line 105, Page 5. Since TEM images are projection image, the fact that (110) lattice fringe is observed does not mean (110) plan is exposed. This is a very critical point since the structure model used for

theoretical calculations are based on this assumption.

Response: Thank you very much for your valuable suggestion. We carefully re-examined our AC-HAADF-STEM images. As shown in Fig. S4, the (110) lattice plane of rutile TiO₂ can be observed in 0.25Pt/STO, 0.5Pt/STO, and 1.0Pt/STO, and this plane is parallel to the interface of the STO support particles. This indicates that the (110) plane is one of the major exposed facets of the STO support. In addition, many relevant studies (*Nat Commun* 2020, 11, 1062; *J Am Chem Soc* 2013, 135, 10673) have reported that the (110) plane is the predominant exposed facet of rutile TiO₂ and have constructed DFT models based on this plane, which is consistent with our findings.

The results in Fig. S3 show that PtO in 1.0Pt/STO exists as nanoparticles, where the observed (110) plane is also parallel to the particle surface. Therefore, we consider the (110) plane to be one of the major exposed facets of PtO and accordingly constructed the DFT model. We fully agree that the observation of a (110) lattice fringe does not necessarily confirm that the (110) plane is exposed. Thus, we have revised the analysis of exposed facets in the manuscript to avoid misunderstanding.

Corresponding text in revised Manuscript:

Page S2:

Supplementary Note 1. The exposed crystal facets of Pt/STO catalysts

The results in Fig. 1b indicate that Pt in the 1.0Pt/STO catalyst primarily exists in the form of nanocrystals. Therefore, the most pronounced Pt aggregation site in the 1.0Pt/STO catalyst was selected to observe the exposed crystal facets of the nanocrystal Pt. As shown in Supplementary Fig. S3, the original AC-HAADF-STEM image was processed by lowering the γ value to identify the brightest region in the processed image, representing the area with the most significant Pt aggregation (highlighted with an orange rectangle). Upon magnification of this region, the interplanar spacing of the nanocrystal Pt was measured to be approximately 0.218 nm, corresponding to the (110)

crystal plane of PtO (JCPDS: #43-1100). The observed (110) plane is parallel to the particle surface, suggesting that it could represent one of the major exposed facets of PtO.

Supplementary Fig. S4 shows that the support structures of all Pt/STO catalysts are rutile-phase TiO₂, with the primary exposed crystal planes being (110), ($\bar{1}10$), and (110). Since rutile TiO₂ belongs to the tetragonal crystal system, these planes are equivalent, indicating that the STO support in all Pt/STO catalysts predominantly exposes the (110) crystal plane. In 0.25Pt/STO, 0.5Pt/STO, and 1.0Pt/STO, this plane is parallel to the interface of the STO support particles. This indicates that the (110) plane is one of the major exposed facets of the STO support. The study of these exposed crystal planes provides theoretical guidance for constructing structural models used in subsequent DFT simulations.

Corresponding figure in revised Manuscript:

Page S8:

Supplementary Fig. S3 Exposed crystal facet of crystalline PtO in 1.0Pt/STO catalyst over AC-HAADF-STEM images.

Supplementary Fig. S4 Exposed crystal facets of STO support of Pt/STO catalysts over AC-HAADF-STEM images.

3. Although the bright spots in the HAADF images are assigned to Sn, and Pt, this interpretation should be approached with caution. Given that atoms are aligned in the depth direction and their positions can affect the image contrast, it is not possible to reliably assign elements based solely on brightness in HAADF images.

Response: In Fig. 1b, assigning the bright spots directly to Sn or Pt solely based on brightness indeed raises concerns regarding reliability. Therefore, we have revised the attribution of the bright spots in Fig. 1b to Pt(Sn), indicating that they may correspond to either Pt or Sn. The corresponding description in the main text has also been modified accordingly.

Corresponding text in revised Manuscript:

Page 5:

As shown in the AC-HAADF-STEM images (Fig. 1b), Pt species evolve with increasing loading: 0.1Pt/STO contains mainly single atoms; 0.25Pt/STO features both single atoms and nanoclusters; 0.5Pt/STO is dominated by nanoclusters; and 1.0Pt/STO exhibits aggregated nanocrystals. Additionally, the intensity analysis of AC-STEM HAADF images reveals a high degree of overlap between Pt and Sn sites, indicating

that Pt has a preferential affinity to Sn sites.

Corresponding figure in revised Manuscript:

Page 6:

Fig. 1| Structural characterization of Pt/STO catalysts. b AC-HAADF-STEM images; the inset shows intensity analysis of the corresponding-colored region; red circle indicates single-atom Pt, yellow circle indicates nanocluster and nanocrystal Pt.

4. The reviewer could not understand the structural model of Pt_{cry}/STO . It appears that a PtO layer is deposited on the STO surface. Is this interpretation correct? Additionally, the reaction seems to proceed exclusively on the PtO layer in a L-H manner. If so, is it reasonable to neglect the potential effect of the Pt-STO interface or the periphery sites?

Response: We first constructed the (110) surface of rutile TiO_2 , and then randomly substituted 20% of Ti atoms with Sn to build the STO structural model. Next, the (110) surface of PtO was generated, and its lattice parameters were matched with those of STO through rotation and cell expansion. The (110) surface of PtO was then placed on top of the STO structure to form a Pt_{cry}/STO heterojunction model. This approach is a

commonly used method to investigate the effect of interfacial interactions between two crystalline materials on their properties. For example, in *Adv Funct Mater* 2025, 35, 2504553, a DFT model was constructed by loading ZnS on a CoS layer (see the Figure a below). Similarly, in *Appl Catal B* 2025, 365, 124864, DFT model was developed by depositing CeO₂ layer on the surface of Co₃O₄ (see the Figure b below).

Figure (for review only) (a) DFT model of ZnS/CoS (*Adv Funct Mater* 2025, 35, 2504553). (b) DFT model of CeO₂/Co₃O₄ (*Appl Catal B* 2025, 365, 124864).

The structural model of Pt_{cry}/STO was constructed primarily to investigate the interfacial interaction between Pt and STO in CO oxidation. To this end, we specifically selected Pt–O–Sn(Ti) sites in Pt_{cry}/STO that are analogous to those in Pt_{sin}/STO and Pt_{clu}/STO as the reaction sites (see Fig. 5a–c). However, during unbiased AIMD simulations after CO adsorption, spontaneous separation between Pt and STO was observed (Fig. 5c), indicating that the Pt–STO interfacial interaction vanishes during the CO oxidation process. Consequently, we infer that CO oxidation in Pt_{cry}/STO proceeds mainly via the PtO layer, which explains the lower activity of Pt_{cry}/STO compared with Pt_{clu}/STO. Therefore, we believe it is reasonable to neglect the potential effect of the Pt–STO interface or the periphery sites.

Corresponding figure in revised Manuscript:

Page 14:

Fig. 5| AIMD simulations and Charge density differences of CO adsorption on Pt/STO catalysts with different Pt sizes. a–c Interatomic distances of Pt-O, Ti-O, and Sn-O during AIMD simulations of CO adsorption on (a) Pt_{sin}/STO, (b) Pt_{clu}/STO, and (c) Pt_{cry}/STO.

We sincerely thank you for the valuable comments and constructive suggestions. These insights have greatly helped us to clarify our structural model and improve the overall quality of the manuscript. We truly appreciate your efforts and time devoted to our work.

Reviewer #2 (Remarks to the Author):

General comments: This manuscript reports a study on Pt size-dependent reverse oxygen spillover (ROS) behavior in CO oxidation over Sn-doped rutile TiO₂ (STO) supports. By combining in situ characterizations with AIMD and DFT simulations, the authors claim to uncover a size-specific trend in ROS intensity, suggesting nanocluster Pt (~0.5 wt%) shows the most pronounced ROS effect and the highest intrinsic activity. While the mechanistic framework is conceptually clear and the methodology is generally sound, the novelty of the study is somewhat limited due to its incremental nature relative to the authors' own prior work (Nat. Commun. 2022), which already established the ROS phenomenon in Pt/STO systems. Several issues need to be clarified or improved, particularly in terms of justification of material choice, the true novelty of the size-dependence finding, the applicability of the reaction conditions, and the speculative nature of key mechanistic claims (e.g., in situ generation of active oxygen). Below are detailed comments:

Response: We sincerely appreciate your valuable comments, which have greatly contributed to the improvement and refinement of our manuscript. We have made every effort to revise the manuscript accordingly and hope that the changes meet your approval. Below are our detailed responses to each of your comments.

(1) The authors should clarify the purpose of the H₂ pre-treatment step. Was it intended to remove surface oxygen species, adjust the Pt oxidation state to Pt²⁺, or promote strong metal-support interactions? What was the direct evidence of its effect, and how does it relate to the proposed ROS mechanism?

Response: We sincerely appreciate your suggestion, which helped us identify certain logical flaws in the manuscript. H₂ pretreatment is a crucial step in catalyst preparation and has a significant influence on catalytic activity. In our previous work (Nat Commun 2023, 14, 3477), we provided a detailed discussion of the effect of H₂ reduction on Pt/STO catalysts. As shown in the Figure (a) below, we first examined the effect of reduction temperature on CO oxidation activity, and the results demonstrated that the

optimum activity was achieved at 300 °C. The Figure (b) below further illustrates the evolution of the chemical state of surface Pt species under different reduction temperatures. The original Pt/STO catalyst contained Pt predominantly in the form of Pt⁴⁺ and Pt²⁺. After H₂ reduction at 300 °C, Pt⁴⁺ was reduced to Pt²⁺. With further increases in reduction temperature, Pt²⁺ remained stable and was not reduced to Pt⁰. Therefore, in our subsequent discussions in this work, we treated the surface Pt species as PtO. Considering that the ROS process in this study refers to the migration of lattice oxygen from the support to Pt active sites for CO oxidation, excessively high oxidation states of Pt (e.g., Pt⁴⁺) would significantly inhibit the ROS process. H₂ reduction effectively lowers the oxidation state of Pt species, making ROS more favorable and thereby enhancing CO oxidation activity. Overall, since the role of H₂ pretreatment has already been discussed extensively in our previous work, in the present study we referred to that discussion when describing the effect of H₂ pretreatment. The corresponding revisions are provided below.

Figure (for review only) (a) Effect of different H₂ pretreatment temperatures on steady-state CO oxidation performance over 0.5Pt/STO with reaction feed of 1% CO, 1% O₂, N₂ balance and GHSV of 60 000 ml g_{cat}⁻¹ h⁻¹. (b) Pt 4f XPS spectra of 0.5Pt/STO before and after pretreatment with H₂ at different temperatures. (*Nat Commun* 2023, 14, 3477)

Corresponding text in revised Manuscript:

Page 19:

After calcination, the materials were reduced in 5% H₂/N₂ at 300 °C for 1 h. The primary purpose of this treatment was to lower the valence state of surface Pt, thereby facilitating the ROS effect and enhancing the CO oxidation activity of the catalyst¹⁹.

(2) The manuscript would benefit from a more detailed justification for using Sn-doped rutile TiO₂. While the previous study introduced the Sn-induced asymmetric Ti-O-Sn sites, it remains unclear whether rutile TiO₂ is chosen for its inherent properties (e.g., stability, crystal facet exposure) or solely for consistency. Were other TiO₂ polymorphs or doping elements considered?

Response: This is a highly meaningful question. First, regarding why we focused on rutile TiO₂ rather than other polymorphs: At the beginning of our ROS studies, we initially selected anatase TiO₂, which is more commonly used in catalytic materials. However, as reported in our previous work (Nat Commun 2023, 14, 3477), when using identical synthesis conditions with only Ti precursors, the resulting TiO₂ was anatase. In contrast, when both Sn and Ti precursors were introduced, the resulting material was Sn-doped rutile TiO₂. Thus, rutile TiO₂ was not deliberately chosen; rather, Sn doping promoted the transformation of TiO₂ into the more thermodynamically stable rutile phase. We have added the relevant clarification in the last paragraph of the Introduction.

Regarding the consideration of other dopants, we believe that ROS is not an incidental phenomenon specific to a single system but a general process that should also occur in systems with similar properties. Therefore, in our future work, we plan to systematically investigate the effect of different dopants—including transition metals and nonmetals—on ROS. In addition, we will explore the influence of different supports such as CeO₂ and Al₂O₃. These studies will enable us to gain deeper insights into the key properties governing ROS. A description of this future research outlook has been added to the final paragraph of the manuscript.

Corresponding text in revised Manuscript:

Page 4:

In this paper, Sn was doped into TiO₂ to synthesize a Ti-based support with asymmetric interfacial oxygen sites. The incorporation of Sn facilitated the phase transformation of TiO₂ into the thermodynamically more stable rutile structure.

Page 19:

We believe that ROS is not an incidental phenomenon confined to a single system but rather a general process that should also occur in systems with similar properties. Therefore, a systematic investigation is needed to examine the effects of different dopants—including transition metals and nonmetals—on ROS. Moreover, the role of different supports (such as CeO₂ and Al₂O₃) should also be explored. These studies will not only deepen the understanding of the fundamental factors governing ROS, but also offer valuable guidelines for the rational design of advanced catalysts with enhanced performance.

(3) Given the authors' prior work in Nat. Commun. already demonstrated the ROS effect on Pt/STO systems, this manuscript appears to focus mainly on Pt size effects, an expected variable in catalysis. The authors need to better emphasize what fundamentally new insight this study provides beyond confirming that nanoclusters show better performance (a commonly observed trend in Pt-based catalysis for CO oxidation).

Response: We sincerely appreciate your suggestion. In response, we revised the descriptions in both the abstract and the activity evaluation section concerning the statement that “nanoclusters show better performance.” Specifically, we reduced the emphasis on the activity of nanoclusters and instead highlighted that the primary focus of this work is the effect of Pt particle size on ROS. The revised text is provided below.

Corresponding text in revised Manuscript:

Page 2:

Pt_{clu} exhibited the most pronounced ROS and thus achieved the highest turnover frequency (TOF) in CO oxidation. The most pronounced ROS was mainly due to the

strongest electron transfer to the interfacial lattice oxygen triggered by CO adsorption with moderate adsorption energy. In contrast, CO adsorption on Pt_{sin} was too strong to initiate ROS, while on Pt_{cry}, it led to a weakening of the interaction between Pt sites and the support, thus hindered the ROS.

Page 11-12:

As shown in Fig. 4a, the 0.1Pt/STO catalyst, on which Pt in the form of single-atom with the highest dispersion, displayed the lowest TOF_{Pt}. As the Pt loading increased, Pt gradually aggregated into nanocluster (0.5Pt/STO), resulting in a TOF_{Pt} 2-5 times higher than that of single-atom Pt. Further increasing Pt content led to the formation of nanocrystal (1.0Pt/STO), with a TOF_{Pt} slightly higher than that of single-atom Pt. There is a remarkable parallelism between the trend of TOF_{Pt} and that of ROS intensity on the Pt/STO catalysts. Specifically, single-atom Pt on 0.1Pt/STO catalyst exhibits scarcely any ROS effect and the lowest TOF_{Pt} value, whereas nanocluster Pt displays the most significant ROS effect and the highest TOF_{Pt}, outperforming both single-atom and nanocrystal Pt on Pt/STO catalyst. This finding strongly suggests that ROS exert a substantial influence on the CO oxidation activity of Pt/STO catalysts.

(4) The transition from single atoms to nanocrystals with increasing Pt loading is quite expected. The corresponding decrease in Pt dispersion and its impact on CO oxidation activity is also well-documented. The authors should be cautious not to overstate this as a novel finding and should instead focus on the mechanistic details of ROS involvement that may be size-specific.

Response: We revisited our description of Pt dispersion and revised the corresponding sections on material characterization. Specifically, we removed redundant statements such as “As the Pt loading increases from 0.1 wt% to 1.0 wt%, the chemical state of Pt on the surface of Pt/STO catalysts undergoes a transition from single atom to nanocluster, and ultimately to nanocrystal,” and streamlined related content. The discussion is now more concise and centers on the core innovation of this work: the Pt size-dependent ROS effect.

Corresponding text in revised Manuscript:

Page 5:

As shown in the AC-HAADF-STEM images (Fig. 1b), Pt species evolve with increasing loading: 0.1Pt/STO contains mainly single atoms; 0.25Pt/STO features both single atoms and nanoclusters; 0.5Pt/STO is dominated by nanoclusters; and 1.0Pt/STO exhibits aggregated nanocrystals.

(5) Since $\text{Ti}(\text{SO}_4)_2$ and SnCl_4 were used as precursors, residual S or Cl species could be present on the STO surface, potentially affecting Pt dispersion, oxidation state, or ROS. The authors should clarify whether these elements were detected (via XPS, ICP-OES, etc.) and whether any such residues impact the observed catalytic behavior.

Response: We conducted XPS analysis on all Pt/STO samples to examine potential S and Cl residues. As shown in Fig. S17, no detectable S or Cl signals were observed in any of the samples. This indicates that the washing process during synthesis effectively removed S and Cl, and their residual influence on the present study can be excluded. The corresponding description has been added to the Synthesis of materials section, as follows.

Corresponding text in revised Manuscript:

Page 19-20:

Since the precursors used for catalyst synthesis contained S and Cl elements, their possible residues could affect the catalytic performance. Therefore, XPS was employed to examine the residual S and Cl species of all Pt/STO catalysts. As shown in Supplementary Fig. S17, no S or Cl signals were detected on the surfaces of any Pt/STO catalysts, indicating that the influence of residual S and Cl on the catalysts can be considered negligible.

Corresponding figure in revised Manuscript:

Page S14:

Supplementary Fig. S17 S 2p and Cl 2p XPS spectra of Pt/STO catalysts.

(6) Line 156 speculates that in situ-generated active oxygen species cannot be detected by O₂-TPD. However, this lacks direct experimental validation. Isotopic labeling experiments (e.g., ¹⁸O₂ vs. ¹⁶O₂) would be highly informative to confirm oxygen migration pathways. The claim of in situ-generated ROS would be far more compelling with such evidence.

Response: Given that the *in situ* active oxygen on the catalyst surface originates from surface lattice oxygen, all Pt/STO catalysts were treated with ¹⁸O₂ at 500 °C for 1 h to partially substitute the surface oxygen species with ¹⁸O. A transient CO oxidation reaction was then carried out at 100 °C. The results show that the formation sequence of C¹⁶O¹⁶O and C¹⁶O¹⁸O follows the order 0.5Pt/STO > 1.0Pt/STO > 0.25Pt/STO > 0.1Pt/STO, which is consistent with our previous transient experiments but differs from the O₂ desorption sequence in O₂-TPD. This further demonstrates that part of the active oxygen involved in CO oxidation over Pt/STO catalysts is generated *in situ*. Moreover, the negligible formation of C¹⁸O¹⁸O indicates that CO reacts with a single oxygen atom on the catalyst surface to produce CO₂. The detailed description of the isotope experiments is provided below.

Corresponding text in revised Manuscript:

Page 7:

The experimental results (Fig. 1e) indicate that the trend in active lattice oxygen content is as follows: 0.5Pt/STO > 1.0Pt/STO > 0.25Pt/STO > 0.1Pt/STO. To further

verify this conclusion, all Pt/STO catalysts were treated with $^{18}\text{O}_2$ at 500 °C for 1 h to partially substitute the surface active oxygen species with ^{18}O . Subsequently, the transition CO oxidation reaction was carried out again at 100 °C. As shown in Supplementary Fig. S7, the formation sequence of $\text{C}^{16}\text{O}^{16}\text{O}$ and $\text{C}^{16}\text{O}^{18}\text{O}$ is consistent with the above conclusion, while almost no $\text{C}^{18}\text{O}^{18}\text{O}$ was detected.

Corresponding figure in revised Manuscript:

Page S9:

Supplementary Fig. S7 Mass spectrometry signals of $\text{C}^{16}\text{O}^{16}\text{O}$, $\text{C}^{16}\text{O}^{18}\text{O}$ and $\text{C}^{18}\text{O}^{18}\text{O}$ produced during the flow of 1% CO/N₂ at 100 °C over Pt/STO catalysts pretreated with $^{18}\text{O}_2$ at 500 °C for 1 h.

(7) The CO oxidation tests were performed at relatively low gas hourly space velocities (GHSV), which are far from realistic exhaust conditions. Furthermore, only one cycle was shown per catalyst, with no assessment of durability or possible structural evolution (e.g., Pt sintering or migration). This limits the practical relevance of the findings.

Response: We greatly appreciate your suggestion regarding the practical application potential of the catalyst. Following your advice, the 0.5Pt/STO sample was selected for activity tests under varying gas hourly space velocities (GHSV), multiple reaction

cycles, and a 24 h stability. As shown in Fig. S11, 0.5Pt/STO exhibits excellent tolerance to high space velocities. Its activity remained nearly unchanged when the GHSV increased from 30,000 to 120,000 mL g_{cat}⁻¹ h⁻¹, and only a slight decline was observed at 240,000 mL g_{cat}⁻¹ h⁻¹. In addition, the catalytic activity was well maintained during four reaction cycles and in the 24 h stability test, indicating that Pt/STO possesses high stability for CO oxidation and strong potential for practical application.

Corresponding text in revised Manuscript:

Page 13:

To evaluate the practical applicability of Pt/STO catalysts, the 0.5Pt/STO sample was selected to activity tests under different gas hourly space velocities (GHSV), multiple reaction cycles, and a 24 h stability. As shown in Supplementary Fig. S11, 0.5Pt/STO exhibits excellent tolerance to high space velocities. Its activity remained nearly unchanged when the GHSV increased from 30,000 to 120,000 mL g_{cat}⁻¹ h⁻¹, and only a slight decline was observed at 240,000 mL g_{cat}⁻¹ h⁻¹. In addition, the catalytic activity was well maintained during four reaction cycles and in the 24 h stability test, indicating that Pt/STO possesses high stability for CO oxidation and strong potential for practical application.

Corresponding figure in revised Manuscript:

Page S11:

Supplementary Fig. S11 (a) CO oxidation activity of 0.5Pt/STO at different GHSV. (b) Cycling performance of CO oxidation over 0.5Pt/STO. (c) 24 h stability of CO oxidation at 110 °C over 0.5Pt/STO.

(8) Realistic exhaust or flue gas streams contain water vapor and often sulfur-containing species. The study lacks any examination of catalyst performance under such conditions. What is the target application of this catalyst? If the study is primarily mechanistic, this should be made explicit, and future studies should be planned accordingly.

Response: We apologize for any unclear descriptions in the manuscript that may have led to a misunderstanding of the primary research focus. As noted, this work is primarily mechanistic, with a focus on the effect of Pt particle size on the ROS effect; therefore, experimental studies and discussion related to practical applications are not taken into consideration. This is because the sulfur and water tolerance of the catalysts were already investigated in our previous work (Nat Commun 2023, 14, 3477). Following the USDRIVE's protocol, we introduced 10% H₂O and 100 ppm SO₂ into the reaction atmosphere and conducted a 7-day stability test. As shown in Figure 1 (for review only), Sn doping significantly enhanced the catalyst's sulfur and water tolerance, indicating substantial practical application potential. In fact, we have already performed pilot-

scale production of Pt/STO catalysts and tested them in industrial furnace flue gas. The results, shown in Figure 2 (for review only), demonstrate that Pt/STO maintains excellent CO oxidation performance under real industrial conditions. In summary, we have revised the Introduction section to clarify the main research objective and highlight the potential application scenarios of the catalyst.

Figure 1 (for review only) Long term test of sulfur resistance in CO oxidation. Reaction conditions: reaction feed of 1% CO, 1% O₂, 10% H₂O, 100 ppm SO₂, N₂ balance; GHSV of 60 000 ml·g_{cat}⁻¹·h⁻¹; and reaction temperature of 240 °C. (Nat Commun 2023, 14, 3477)

Figure 2 (for review only) CO and C₇H₈ oxidation performance of Pt/STO in a pilot-

scale coking flue gas.

Corresponding text in revised Manuscript:

Page 4:

Consequently, nanocluster Pt exhibited a pronounced ROS effect, enhancing CO catalytic oxidation and achieving a turnover frequency (TOF) approximately twice that of the single-atom and nanocrystal Pt. These findings provide deeper insight into the interfacial oxygen migration mechanism and offer a theoretical basis for designing catalysts with ROS-enhanced activity for CO removal in industrial furnace flue gas.

(9) The current findings may be specific to Sn-doped rutile TiO₂ systems. The authors should discuss whether the ROS mechanism observed here could be extrapolated to other supports or dopants. If it is a highly system-specific effect, its broader catalytic relevance is limited.

Response: We greatly appreciate your suggestion. Following your this and the second suggestions, we have added a section on future research directions at the end of Discussion. We believe that ROS is not an incidental phenomenon limited to a single system, but rather a general process likely to occur in systems with similar properties. Therefore, the effects of various factors, including different dopants and supports, should be systematically investigated to identify the key determinants governing ROS.

Corresponding text in revised Manuscript:

Page 19:

We believe that ROS is not an incidental phenomenon confined to a single system but rather a general process that should also occur in systems with similar properties. Therefore, a systematic investigation is needed to examine the effects of different dopants—including transition metals and nonmetals—on ROS. Moreover, the role of different supports (such as CeO₂ and Al₂O₃) should also be explored. These studies will not only deepen the understanding of the fundamental factors governing ROS, but also

offer valuable guidelines for the rational design of advanced catalysts with enhanced performance.

(10) Theoretical models were based on as-prepared catalysts, yet ROS and particle restructuring could occur during catalytic testing. Therefore, the models may not reflect the true stabilized catalytic surfaces. The authors should comment on this limitation and whether alternative models (e.g., post-reaction structures) were considered.

Response: Our theoretical models were first subjected to AIMD simulations at 700 K, after which five representative configurations from the simulation were selected for structural optimization. The lowest-energy configuration was then used for further theoretical calculations (detailed modeling procedures are provided in Supplementary Note 4). Thus, the influence of particle restructuring was, to some extent, already accounted for through the high-temperature AIMD simulations. Moreover, regardless of whether the reaction involves ROS, our simulations included the complete reaction cycle (see Fig. 6), ensuring that all catalyst models returned to their initial state after CO oxidation. Consistent with this, our stability experiments (Fig. S11) confirmed that the catalysts exhibit good durability. Therefore, we believe that the theoretical models in this work represent the stabilized catalytic surfaces when investigating the size-dependent effect of Pt on ROS, which is the central focus of this study. We fully acknowledge that prolonged reactions to industrial-scale durations (e.g. 8000 hours) may induce surface changes in the catalysts, but catalyst aging lies beyond the scope of this work and will be considered in future studies on ROS effects.

Corresponding figure in revised Manuscript:

Page 17:

Fig. 6 | CO oxidation reaction cycles on Pt/STO catalysts with different Pt sizes. **a–c** CO oxidation reaction cycles and energy profiles for (a) Pt_{sin}/STO, (b) Pt_{clu}/STO, and (c) Pt_{cry}/STO. **d** Schematic illustration of the CO oxidation mechanism on PtO clusters via a reverse oxygen spillover pathway to form CO₂ on the Pt/STO catalyst surface.

Page S11:

Supplementary Fig. S11 (a) CO oxidation activity of 0.5Pt/STO at different GHSV. (b)

Cycling performance of CO oxidation over 0.5Pt/STO. (c) 24 h stability of CO oxidation at 110 ° C over 0.5Pt/STO.

We sincerely thank you for the valuable comments and constructive suggestions. These insights have greatly helped us to clarify the application potential and improve the overall quality of the manuscript. We truly appreciate your efforts and time devoted to our work.

Detailed Response to Reviewers

Response to Reviewer #1:

I think the authors have revised the manuscript satisfactorily except for the reviewer's second comment. The authors insist that (110) plane is parallel to the particle surface and/or Pt/STO interface, but the reviewer does not recognize surface nor interface in the ADF images. Where are they? The authors seem not understand the reviewer's comment that "TEM images are projection images". It is very difficult to believe that main exposed facet of the STO nanoparticles with indefinite structure is (110) at least from the AC-HAADF-STEM images shown.

Response: We appreciate your insightful comment regarding the interpretation of TEM projection images. We fully agree that TEM images are 2D projections of 3D structures, which can make the identification of specific surface facets challenging. To address this, we have now added explicit labels to Supplementary Figures S3 and S4 to clearly indicate the identified catalyst surfaces.

While acknowledging the projection nature of STEM, we determined the exposed facets by identifying the lattice planes parallel to the particle boundaries under specific zone axes. The measured lattice fringe spacings and the corresponding FFT patterns consistently point to the (110) plane as the primary exposure in these regions.

Supplementary Fig. S3 Exposed crystal facet of crystalline PtO in 1.0Pt/STO catalyst over AC-HAADF-STEM images.

Supplementary Fig. S4 Exposed crystal facets of STO support of Pt/STO catalysts over AC-HAADF-STEM images.

Response to Reviewer #2:

The authors have responded to my previous comments with a level of thoroughness and technical depth that substantially strengthens the manuscript. All major concerns raised regarding novelty, mechanistic interpretation, experimental validation, and practical relevance have been carefully addressed through new experiments, revised analyses, and clearer framing of the study's scope. In particular, the authors have convincingly clarified the conceptual advance of this work relative to their prior publications by shifting the focus from activity trends to the size-dependent modulation of reverse oxygen spillover (ROS) and its mechanistic origin. The revised discussion now articulates how CO adsorption strength, interfacial electron transfer, and Pt structural evolution jointly determine ROS intensity, rather than reiterating expected dispersion-activity relationships. This significantly improves the fundamental insight provided by the study. Several points that were previously speculative are now supported by additional experimental evidence. Notably, the inclusion of ^{18}O isotope-labeling experiments provides direct validation of in situ lattice oxygen participation in CO oxidation, strengthening the central ROS claim beyond indirect inference from O_2 -TPD alone. The added GHSV-dependent activity tests, cycling experiments, and 24 h

stability data also address concerns regarding kinetic relevance and catalyst robustness, demonstrating that the conclusions are not artifacts of a narrow operating window. From a theoretical standpoint, the authors have adequately justified their structural models and clarified the treatment of interfacial effects and catalyst restructuring through AIMD simulations. While long-term catalyst aging remains beyond the scope of the present study, this limitation is now explicitly acknowledged and does not undermine the mechanistic conclusions targeted here. Overall, the revised manuscript presents a coherent and well-supported mechanistic framework for Pt size-dependent ROS behavior. In its current form, the work meets the standards of Nature Communications in rigor, clarity, and conceptual contribution. I therefore recommend acceptance without further revision.

Response: We sincerely thank you for the positive assessment and for recognizing the strengthened mechanistic depth and experimental rigor of our revised manuscript. We are also grateful for your constructive feedback throughout the review process, which has significantly improved the quality of this work.